# Single-Rollout Hidden-State Dynamics for Training-Free RLVR Data Selection

**Jianghao Wu**[1]  **Jianfei Cai**[1]  **Weiqiang Wang**[1]  **Jin Ye**[1]  **Daniel F. Schmidt**[1]  **Yasmeen George**[1]

## Abstract

Reinforcement learning with verifiable rewards (RLVR) can yield large reasoning gains from very few training instances, yet its strong sensitivity to which instances are used makes data selection a central bottleneck. Most existing selection pipelines rely on training-time optimization signals and/or require access to verifiable rewards or ground-truth answers over large candidate pools, which is costly and often infeasible in specialized domains. We study RLVR data selection in a setting where selection must be performed *before* any RL training and *without* labels or reward evaluation on the full pool. We propose **SHIFT**, a one-shot, training-free selector based solely on inference-time hidden-state dynamics. For each candidate instance, SHIFT runs a single deterministic reasoning rollout and computes a *reasoning-induced representation shift* (RIRS) as the start-to-end hidden-state delta. SHIFT uses the RIRS magnitude as a lightweight proxy for instance utility and enforces coverage via a quality-weighted farthest-first CoreSet procedure in an RIRS-augmented feature space, producing compact subsets that scale to large unlabeled pools. Across mathematical reasoning and medical QA benchmarks under ultra-low budgets, SHIFT consistently outperforms training-free diversity and difficulty/uncertainty baselines, improving both in-domain accuracy and transfer to harder evaluation settings. Ablations show that RIRS-based coverage and quality-weighting contribute complementary gains, and analyses indicate that RIRS is not explained by simple input/output length statistics. Code is available at github.com/JianghaoWu/SHIFT.

---

[1]Faculty of Information Technology, Monash University, Melbourne, Australia. Correspondence to: Jianghao Wu <jianghao.wu@monash.edu>, Weiqiang Wang <weiqiang.wang@monash.edu>.

*Proceedings of the $43^{rd}$ International Conference on Machine Learning*, Seoul, South Korea. PMLR 306, 2026. Copyright 2026 by the author(s).

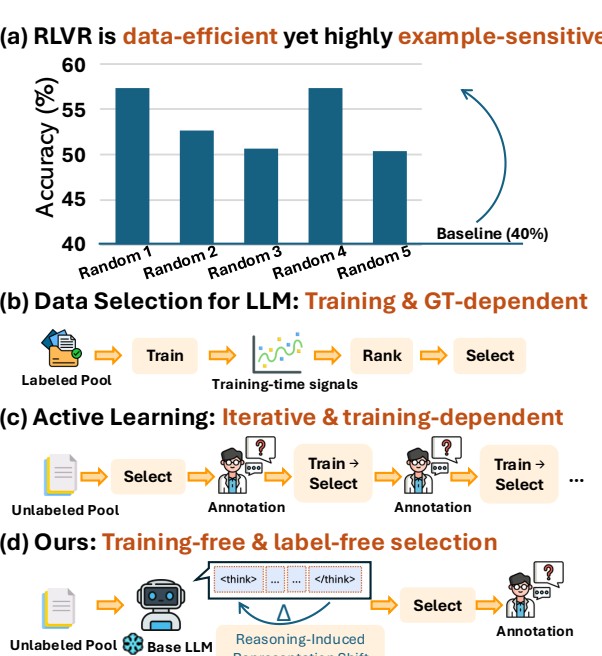

**(a)** RLVR is data-efficient yet highly example-sensitive

**(b)** Data Selection for LLM: Training & GT-dependent

**(c)** Active Learning: Iterative & training-dependent

**(d)** Ours: Training-free & label-free selection

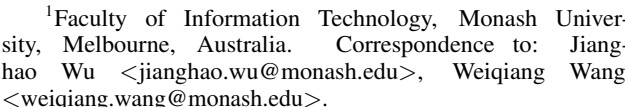

*Figure 1.* RLVR is highly example-sensitive (a), while prior selection methods rely on training-time and/or ground-truth signals (b,c). We enable one-shot, training-free and label-free selection via reasoning-induced representation shifts (d).

## 1. Introduction

Reinforcement learning with verifiable rewards (RLVR) has recently emerged as a powerful paradigm for enhancing the reasoning ability of large language models (LLMs) (Guo et al., 2025; Wang et al., 2025b; Yue et al., 2025; Wen et al., 2025). Recent studies suggest that RLVR can be extremely data-efficient: in some settings, with only a single carefully chosen training example, an LLM may achieve substantial improvements on challenging reasoning benchmarks, sometimes approaching the performance obtained using thousands of RL training instances (Wang et al., 2025c; Li et al., 2026; 2025; Chen et al., 2025). This observation suggests the possibility that strong reasoning gains may be unlocked by RLVR with only a handful of "impactful" examples, potentially making RL adaptation far more accessible than previously believed.

Yet this promise immediately raises a fundamental ques-

tion: how can we efficiently identify impactful RL training examples? Figure 1 illustrates the central challenge and our proposed solution. Prior work shows that one-shot RLVR is highly sensitive to the chosen example (Wang et al., 2025c), making data selection a key bottleneck for practical low-shot RLVR. A representative line of work ranks candidate instances using training-dependent signals, such as reward/accuracy trajectories, gradient-based statistics, or other optimization dynamics observed during (proxy) fine-tuning, and selects those with large fluctuations or high estimated utility (Wang et al., 2025c; Naharas et al., 2025). However, such strategies typically assume (i) access to verifiable rewards, which often requires ground-truth answers, and (ii) running at least partial fine-tuning over a large candidate pool to obtain the ranking signals. In many realistic domains (e.g., medical QA or specialized scientific reasoning), reliable annotations are costly, and performing RL over thousands of candidates merely to decide what to train on is computationally wasteful. As a result, existing RLVR data selection pipelines remain label- and training-dependent, limiting their applicability in truly low-resource settings. Active learning (AL) similarly aims to select a small subset for annotation under a limited budget; however, classical AL criteria typically rely on iterative select–label–train updates or training-time uncertainty/gradient signals (Settles, 2009; Wang et al., 2023; Yuan et al., 2023; Chen et al., 2024), and thus do not directly apply to low-budget RLVR. In particular, RLVR utility is reward-dependent, whereas most AL signals are defined via supervised losses or calibrated predictive distributions that are unavailable prior to annotation.

In this work, we pursue a different route: can we identify a small set of high-impact training instances from a large unlabeled pool, and spend annotation budget only on these selected examples, *without* requiring ground-truth answers, verifiable rewards, or any training-time optimization signals during selection? To this end, we introduce SHIFT, a one-shot, training-free selector that leverages inference-time hidden-state dynamics to estimate which instances are likely to be most useful for downstream RLVR. Our approach is motivated by evidence that reasoning exhibits structured internal dynamics. Recent theory suggests that transformer computation can be interpreted through implicit low-rank update viewpoints, offering a mechanistic account of learning-like behavior during reasoning (Dherin et al., 2025). Meanwhile, empirical studies show that hidden-state trajectories encode informative signatures of reasoning processes, and that simple summaries such as start-to-end hidden-state differences capture non-trivial structure of the computation (Liang et al., 2025). Building on these observations, SHIFT uses the *reasoning-induced representation shift* (RIRS), the start-to-end hidden-state delta under a single deterministic reasoning rollout, as a training-free proxy for an instance's learning potential under RLVR.

Concretely, for each unlabeled instance, SHIFT runs one deterministic reasoning trace, extracts hidden states at the beginning and the end of the trace, and computes RIRS. It then selects a compact subset by jointly optimizing two desiderata: (i) *informativeness*, measured by the RIRS magnitude as a lightweight utility proxy; and (ii) *coverage*, enforced via a farthest-first CoreSet strategy in an RIRS-augmented feature space. Finally, we annotate only the selected instances to obtain verifiable rewards and run RLVR on this small set, substantially reducing reward/annotation and training overhead under severe supervision scarcity.

Our main contributions are as follows.

- We formalize RLVR data selection in a *pre-RL* regime, where selection must be done on a large unlabeled pool *without* reward evaluation or training-time optimization signals, and rewards are obtained only for a small selected subset.

- We propose SHIFT, a training-free one-shot selector that leverages inference-time hidden-state dynamics. SHIFT quantifies instance utility via the *reasoning-induced representation shift* (RIRS) and promotes coverage using an RIRS-augmented feature space.

- We instantiate SHIFT with a *single-rollout* quality-weighted farthest-first CoreSet algorithm that scales to large pools, and demonstrate consistent gains on mathematical reasoning and medical QA under ultra-low budgets, supported by ablations and analyses on proxy validity and compute trade-offs.

## 2. Related Work

### 2.1. Data Selection for LLM Training and Adaptation

Selecting a small yet effective subset of training data has become a central problem for improving the efficiency of LLM post-training. In instruction tuning, recent evidence suggests that a relatively small set of carefully curated instruction–response pairs can already yield strong performance (Zhou et al., 2023). Motivated by this observation, a number of studies explore scalable ways to construct or filter high-quality training data, including using LLMs (e.g., ChatGPT) to synthesize instruction-following data, as well as leveraging educational sources such as textbooks (Eldan & Li, 2023; Gunasekar et al., 2023; Chen et al., 2023). Beyond data synthesis and heuristic filtering, another line of work develops *training-signal-driven* selection criteria based on optimization dynamics. Gradient-based approaches estimate an example's usefulness via statistics of gradients, such as gradient variance (Agarwal et al., 2022; Anand et al., 2023; Wang et al., 2026), or select subsets by matching gradients between a candidate pool and a target objective (Killamsetty

et al., 2021; Xia et al., 2024; Wang et al., 2025a). Difficulty-based criteria have also been studied across different stages of the LLM pipeline. For pre-training, perplexity under a strong reference model has been used as a lightweight proxy to filter data (Marion et al., 2023). For alignment, difficulty can be quantified using reward-based signals, such as the gap between accepted and rejected responses (Qi et al., 2025). Most closely related to our work, Wang et al. (2025c) show that reinforcement learning with verifiable rewards (RLVR) can be remarkably data-efficient for reasoning: even a single carefully chosen training instance may yield substantial gains. They select such impactful examples by ranking candidates with a *Historical Variance Score*, defined as the variance of each sample's training accuracy across RL epochs. While effective, this family of selection strategies typically relies on running training (or RL) over a candidate pool to obtain optimization-time signals, and often assumes access to verifiable rewards or ground-truth answers. These requirements can be prohibitive in practical deployments, especially in specialized domains such as medical reasoning, where annotations are costly and large-scale trial training for data selection is computationally expensive.

### 2.2. Active Learning

Active learning (AL) studies how to obtain supervision as efficiently as possible under a constrained annotation budget, typically by prioritizing unlabeled samples that are expected to be most informative once labeled (Settles, 2009; Ren et al., 2021). Existing AL approaches are often organized into three broad categories. **Uncertainty-based** methods prioritize samples on which the current model is least confident, under the intuition that resolving ambiguous cases yields the largest learning gains. Representative criteria include least-confidence sampling, margin-based sampling (Monarch, 2021), and entropy-based measures (Shannon, 1948), which quantify uncertainty from predicted class probabilities. Beyond direct probability-based scores, some works estimate uncertainty through surrogate objectives, e.g., predicting loss values for candidate instances (Yoo & Kweon, 2019; Huang et al., 2021). Other variants characterize uncertainty via prediction disagreement across multiple stochastic views, such as different data augmentations (Gao et al., 2020), standard versus dropout-enabled inference (Gal & Ghahramani, 2016; Gal et al., 2017), or perturbations in feature space (Parvaneh et al., 2022). **Diversity-based** methods instead aim to select a subset that broadly represents the underlying data distribution, thereby avoiding redundancy and improving coverage. Common strategies include core-set selection (Caramalau et al., 2021; Sener & Savarese, 2017) and clustering-based sampling in a learned embedding space (Kutsuna et al., 2012; Nguyen & Smeulders, 2004; Urner et al., 2013). Related formulations introduce explicit diversity constraints during optimization (Yang et al.,

2015; Elhamifar et al., 2013). **Hybrid** approaches attempt to jointly account for uncertainty and diversity. For instance, (Elhamifar et al., 2013; Chen et al., 2024) apply clustering over gradient embeddings to balance informativeness and coverage. Several works further adopt multi-stage selection pipelines, where candidates are first filtered by one criterion and then refined by another (Parvaneh et al., 2022; Wang et al., 2023; Yuan et al., 2023). Nevertheless, many hybrid designs remain largely driven by aleatoric uncertainty signals, whose adequacy and robustness may be limited in realistic settings. However, most AL methods follow an iterative *select–label–train* loop and rely on training-time signals (e.g., uncertainty, loss/gradient surrogates, or stochastic disagreement) to guide querying. This paradigm can be unreliable or prohibitively expensive for low-budget RLVR, since example utility is reward-dependent and fine-tuning a large unlabeled pool merely to obtain selection signals defeats data-efficiency. Moreover, training-free AL variants often rely on static input representations, which may not capture an LLM's *reasoning-time* computation. Therefore, we focus on *one-shot* selection that is label-free and training-free at selection time, and leverage inference-time hidden-state dynamics as a proxy for RLVR utility.

## 3. Methodology

### 3.1. Problem Formulation

Let $\mathcal{U} = \{x_i\}_{i=1}^N$ denote a large pool of unlabeled reasoning instances (e.g., math or medical questions), and let $B$ be the selection budget. Our goal is to select a compact subset $S \subset \mathcal{U}$ with $|S| = B$ for downstream RLVR training, while ensuring that the selection process itself is *training-free* and *label-free*. That is, the selection procedure must not rely on ground-truth answers, verifiable rewards, or any optimization-time signals.

We assume access to a base language model $f_\theta$ and perform only inference-time computation over $\mathcal{U}$. For each candidate instance $x \in \mathcal{U}$, we generate a single deterministic reasoning trace using greedy decoding (temperature $T = 0$), which yields a stable hidden-state trajectory. We then extract a compact representation of reasoning dynamics and use it to estimate the utility of selecting $x$ for RLVR.

### 3.2. Hidden-State Delta Representation

Given an instance $x \in \mathcal{U}$, we generate a single deterministic reasoning trace $\mathbf{y}(x)$ using greedy decoding ($T = 0$) under a fixed reasoning prompt template. We then identify two anchor token positions corresponding to the beginning and the end of the model's reasoning trace, and extract the hidden states at these two positions. In particular, when the model produces an explicit chain-of-thought (CoT) segment, we take the first and the last tokens of the CoT span as the start

and end anchors. For models that support reasoning delimiters (e.g., `<think>` and `</think>`), these anchors can be instantiated as the hidden states at the delimiter tokens.

Let $\mathbf{h}_t^{(\ell)}(x) \in \mathbb{R}^D$ denote the hidden state at token position $t$ from transformer layer $\ell$ during generation. Following Liang et al. (2025), we aggregate information across all layers by averaging the anchor hidden states:

$$\mathbf{s}(x) = \frac{1}{L_{\text{layer}}} \sum_{\ell=1}^{L_{\text{layer}}} \mathbf{h}_{t_s}^{(\ell)}(x) \in \mathbb{R}^D, \qquad (1)$$

$$\mathbf{e}(x) = \frac{1}{L_{\text{layer}}} \sum_{\ell=1}^{L_{\text{layer}}} \mathbf{h}_{t_e}^{(\ell)}(x) \in \mathbb{R}^D. \qquad (2)$$

We then define the reasoning-induced representation shift (RIRS) as the start-to-end hidden-state delta:

$$\Delta(x) \;=\; \mathbf{e}(x) - \mathbf{s}(x) \;\in\; \mathbb{R}^D. \qquad (3)$$

**Theoretical motivation and proxy justification.** Recent theory (Dherin et al., 2025) offers a mechanistic view of in-context learning in transformer blocks as an *implicit weight-transfer* phenomenon. Consider a transformer block in which a contextual layer (e.g., self-attention) is followed by an MLP. Under this view, the effect of a context portion $Y$ on the block output coincides with the effect of removing $Y$ from the context and instead applying a rank-1 update to the first dense layer of the MLP (Dherin et al., 2025). The weight update admits the explicit form

$$\Delta W(Y) = \frac{\left(W\,\Delta A(Y)\right) A(C \setminus Y, x)^\top}{\|A(C \setminus Y, x)\|_2^2}, \qquad (4)$$

where $\Delta A(Y) = A(C, x) - A(C \setminus Y, x)$ is the context-induced change of the contextual-layer output. Since $\Delta W(Y)$ is rank-1, applying $\|uv^\top\|_F = \|u\|_2\|v\|_2$ together with the sub-multiplicativity of the spectral norm $\|W\|_2$ yields

$$\|\Delta W(Y)\|_F \;\leq\; \frac{\|W\|_2}{\|A(C \setminus Y, x)\|_2}\,\|\Delta A(Y)\|_2, \quad (5)$$

so the magnitude of the implicit update is upper-bounded (up to the data-dependent factor $\|W\|_2/\|A(C \setminus Y, x)\|_2$) by the context-induced representation change $\|\Delta A(Y)\|_2$. This motivates using observable representation shifts as a proxy for an instance's capacity to drive internal adaptation. We stress, however, that Eq. (5) is a block-level statement for a fixed query position, whereas RIRS aggregates information across layers over an entire reasoning trace. We therefore use this theory as motivation rather than derivation: $\|\Delta(x)\|_2$ is an empirically motivated surrogate of net context-induced change along a deterministic rollout, not a formal estimator of $\|\Delta A(Y)\|_2$ or $\|\Delta W(Y)\|_F$.

**From self-generated CoT context to an observable trajectory-level surrogate.** In our setting, the CoT segment between the `<think>` and `</think>` delimiters is *self-generated* and becomes part of the autoregressive context for subsequent tokens. Building on the per-token decomposition implicit in (Dherin et al., 2025), each newly added token in the trace can induce a small context-conditioned implicit update within transformer blocks, so the overall inference process can be interpreted as accumulating such context-induced effects across tokens and layers. However, $\Delta A(Y)$ in Eq. (5) is defined at the output of a single contextual layer and is not directly observable at scale across all blocks without intrusive instrumentation. We therefore use the trajectory-level residual hidden-state shift between start and end anchors, $\Delta(x) = \mathbf{e}(x) - \mathbf{s}(x)$, as a lightweight, layer-aggregated surrogate of the net context-induced representation change along the deterministic reasoning rollout. Accordingly, we use $\|\Delta(x)\|_2$ as a training-free proxy for how strongly an instance engages the model's internal computation. We hypothesize that this quantity correlates with the instance's utility for downstream RLVR, and empirically verify this in Section 5. We do not claim $\Delta(x)$ equals $\Delta A(Y)$ or $\Delta W(Y)$ for any particular block; rather, $\|\Delta(x)\|_2$ is an observable summary motivated by the control relationship in Eq. (5) and the autoregressive structure of self-generated CoT reasoning.

### 3.3. Utility Score from Hidden-State Delta

Based on the reasoning-induced representation shift (RIRS) $\Delta(x)$ defined in Eq. (3), we define a training-free utility score as the $\ell_2$ norm of the start-to-end hidden-state delta:

$$q(x) \;=\; \|\Delta(x)\|_2. \qquad (6)$$

Intuitively, $q(x)$ measures the strength of the model's internal representation change induced by a single deterministic reasoning trace, and we use it as a lightweight proxy for the instance utility in downstream RLVR.

To reduce scale variation across instances and improve numerical stability, we apply a monotonic log transform:

$$\tilde{q}(x) \;=\; \log(1 + q(x)). \qquad (7)$$

We use $\tilde{q}(x)$ as the final utility score for subset selection.

### 3.4. Coverage Features for Diversity

Selecting only the highest-utility instances may lead to redundant examples that occupy a narrow region of the feature space. To encourage diversity and improve coverage, we construct a feature representation $\phi(x)$ for each instance and perform CoreSet-style selection in this space.

Specifically, we represent each instance by concatenating its start-state representation and reasoning-induced hidden-

state delta:

$$\phi(x) \ = \ \frac{\left[ \ \mathbf{s}(x) \ ; \ \Delta(x) \ \right]}{\left\| \left[ \ \mathbf{s}(x) \ ; \ \Delta(x) \ \right] \right\|_2} \ \in \ \mathbb{R}^{2D}, \qquad (8)$$

where $[\cdot \, ; \cdot]$ denotes vector concatenation and $\phi(x)$ is $\ell_2$-normalized so that all coverage distances are computed on the unit sphere. This feature design captures both (i) the instance-conditioned representation at the start of reasoning via $\mathbf{s}(x)$ and (ii) the reasoning-induced transformation via $\Delta(x)$, enabling diversity selection that accounts for both instance context and reasoning dynamics.

### 3.5. Quality-Weighted Farthest-First Selection

We now describe our final selection algorithm, which balances utility and diversity in a single greedy procedure. Let $S$ be the current selected set. For each candidate instance $x \in \mathcal{U} \setminus S$, we define its coverage distance to the selected set as

$$d(x, S) \ = \ \min_{x' \in S} \| \phi(x) - \phi(x') \|_2 . \qquad (9)$$

We initialize the selected set with the highest-utility instance:

$$S \leftarrow \left\{ \arg \max_{x \in \mathcal{U}} \tilde{q}(x) \right\}. \qquad (10)$$

Then, we iteratively add instances according to a quality-weighted farthest-first rule:

$$x^\star \ = \ \arg \max_{x \in \mathcal{U} \setminus S} \tilde{q}(x) \cdot d(x, S). \qquad (11)$$

We repeat this procedure until $|S| = B$.

This greedy strategy prioritizes instances that are simultaneously high-utility (large $\tilde{q}(x)$) and complementary to previously selected examples (large $d(x, S)$), producing a compact subset suitable for efficient RLVR adaptation.

## 4. Experiments

### 4.1. Experimental Setup

**Models.** We evaluate SHIFT on a compact yet representative set of LLMs. Specifically, we use Qwen3-1.7B (Yang et al., 2025) for the medical QA tasks and the math-specialized Qwen2.5-Math-1.5B (Yang et al., 2024) for mathematical reasoning. All experiments are initialized from publicly released checkpoints.

**Benchmarks.** We evaluate SHIFT on two task families: medical question answering and mathematical reasoning. For medical QA, we adopt MEDQA (Jin et al., 2021), a multiple-choice dataset constructed from United States Medical Licensing Examination (USMLE) questions. We use the official split with 10.2K training examples as the unlabeled

---

**Algorithm 1** One-shot Training-free and Label-free RLVR Data Selection via RIRS

**Input:** Unlabeled pool $\mathcal{U} = \{x_i\}_{i=1}^{N}$; budget $B$; base LM $f_\theta$; prompt template $\mathcal{P}$; layer count $L_{\text{layer}}$.
**Output:** Selected subset $S \subset \mathcal{U}$ with $|S| = B$.

**foreach** $x \in \mathcal{U}$ **do**

    Generate a deterministic reasoning trace $\mathbf{y}(x)$ using $f_\theta$ with greedy decoding $(T = 0)$ under $\mathcal{P}$

    Identify start/end anchor indices $(t_s, t_e)$ of the reasoning trace

    $\mathbf{s}(x) \leftarrow \frac{1}{L_{\text{layer}}} \sum_{\ell=1}^{L_{\text{layer}}} \mathbf{h}_{t_s}^{(\ell)}(x)$

    $\mathbf{e}(x) \leftarrow \frac{1}{L_{\text{layer}}} \sum_{\ell=1}^{L_{\text{layer}}} \mathbf{h}_{t_e}^{(\ell)}(x)$

    $\Delta(x) \leftarrow \mathbf{e}(x) - \mathbf{s}(x)$

    $q(x) \leftarrow \|\Delta(x)\|_2$

    $\tilde{q}(x) \leftarrow \log(1 + q(x))$

    $\phi(x) \leftarrow [\mathbf{s}(x); \Delta(x)]$

    $\phi(x) \leftarrow \phi(x) / \|\phi(x)\|_2$

$S \leftarrow \{\arg \max_{x \in \mathcal{U}} \tilde{q}(x)\}$

**while** $|S| < B$ **do**

    **foreach** $x \in \mathcal{U} \setminus S$ **do**

        $d(x, S) \leftarrow \min_{x' \in S} \|\phi(x) - \phi(x')\|_2$

        $\text{score}(x) \leftarrow \tilde{q}(x) \cdot d(x, S)$

    $x^\star \leftarrow \arg \max_{x \in \mathcal{U} \setminus S} \text{score}(x)$

    $S \leftarrow S \cup \{x^\star\}$

**return** $S$

---

pool for data selection, and report accuracy on the 1.27K test set. To assess cross-dataset generalization, we further evaluate the resulting models on MEDMCQA (Pal et al., 2022), PUBMEDQA (Jin et al., 2019), and MedXpertQA (Zuo et al., 2025). In particular, MedXpertQA consists of two subsets targeting medical understanding (U) and clinical reasoning (R), respectively. For mathematical reasoning, we use MATH-500 (Hendrycks et al., 2021), which covers seven subject categories: Algebra (Alg.), Counting & Probability (C. P.), Geometry (Geo.), Intermediate Algebra (I. Alg.), Number Theory (N. T.), Prealgebra (Prealg.), and Precalculus (Precal.). We use 350 examples for data selection and evaluate on the remaining 150 examples, and additionally report results on the American Mathematics Competitions (AMC) benchmark to measure out-of-distribution generalization. We consider small selection budgets to evaluate SHIFT under limited supervision: for MEDQA, we select $K \in \{10, 20\}$ examples from the unlabeled pool (0.1% and 0.2% of the training set), while for MATH-500, we select $K \in \{7, 14\}$ examples from the 350-example selection split (2% and 4% of the pool).

**Comparison Methods.** We compare SHIFT with training-free selection baselines spanning (i) diversity/coverage sampling and (ii) difficulty/uncertainty heuristics. As reference points, we report an oracle *Full-Data* reference (RLVR

on the full pool) and *Random* sampling (mean over five seeds). We use a fixed question embedding space given by `sentence-transformers/all-MiniLM-L6-v2` for the diversity baselines: *KMeans-Center* (**Cluster**) selects the instance closest to each $k$-means centroid, and *Farthest-First* (**CoreSet** (Sener & Savarese, 2017)) greedily selects instances that maximize the minimum distance to the current selected set. For difficulty, we include *Q-PPL* (select the $K$ questions with the highest prompt perplexity under the base model). For self-consistency uncertainty, we run $R$=32 stochastic rollouts per instance with temperature $T$=0.6. *SC-Entropy* ranks instances by the entropy of the empirical answer distribution across rollouts (higher is more uncertain). *CoT Similarity* encodes each sampled chain-of-thought with `all-MiniLM-L6-v2`, computes the average pairwise cosine similarity among the $R$ sampled CoTs, and selects instances with *lower* similarity (i.e., less consistent reasoning). Finally, *A-PPL* ranks instances by the perplexity of the model-generated answer (conditioned on the prompt) under the base model. All methods select $K$ instances from the same unlabeled pool and then perform RLVR with identical training budgets and hyperparameters; only the selection rule differs.

**Implementation Details and Evaluation Protocol.** We implement SHIFT with Group Relative Policy Optimization (GRPO) on all benchmarks. For each training question, we sample $N$=8 rollouts using temperature 0.7 and top-$p$=0.95, compute verifiable rewards with task-specific answer checkers, and update the policy via GRPO with a KL regularization term anchoring the policy to the base model. Unless otherwise stated, we use a unified optimization setup (optimizer, learning rate schedule, batch settings, and reward normalization) across all methods, and **only** change the data selection strategy for fair comparison.

At evaluation, we use greedy decoding (temperature 0) and report Pass@1 accuracy. To ensure consistent grading across datasets and output formats, we apply a standardized answer normalization and equivalence-checking pipeline (including text canonicalization and symbolic matching via `sympy` when applicable); full details follow our `grade_answer` implementation in Appendix B. We cap the maximum generation length at 3072 tokens for both training rollouts and evaluation. Unless otherwise stated, training/evaluation uses a fixed seed; Random sampling is reported as the mean over five selection seeds. All experiments run on $8\times$ NVIDIA A100 80GB GPUs. We use the EasyR1 framework (Zheng et al., 2025) with a unified configuration to control for implementation differences. Detailed hyperparameters and training configurations are provided in Appendix C.

## 4.2. Main Results

**Results on Mathematical Reasoning.** Table 1 summarizes RLVR under extremely small budgets (2% and 4% selected from MATH-500), evaluated on both MATH-500 (in-domain) and AMC (OOD). Across budgets, SHIFT delivers the strongest or near-strongest performance among training-free baselines, with particularly large gains in the ultra-low (2%) regime. With only 2% selected examples, SHIFT achieves **62.67** on MATH-500 and **38.55** on AMC, improving over *Random* by +8.94 / +12.77 (53.73 / 25.78) and outperforming standard diversity-only methods such as *Cluster* (44.67 / 25.30) and *CoreSet* (47.33 / 25.30). It also surpasses difficulty/uncertainty heuristics, including *Q-PPL* (60.00 / 30.12), *A-PPL* (55.33 / 31.33), and *SC-Entropy* (56.00 / 31.33), indicating that reasoning-induced hidden-state dynamics provide a more faithful training-free proxy for RLVR utility than surface-level difficulty or self-consistency uncertainty alone. Notably, despite using only 2% data, SHIFT even exceeds the full-data RLVR reference on AMC (38.55 vs. 33.73), suggesting that selecting high-impact examples can improve OOD generalization beyond simply increasing the training pool. Category-wise, SHIFT yields broad gains on MATH-500 at 2% compared to *Random*, e.g., Algebra (77.78 vs. 69.44), Counting & Probability (58.33 vs. 45.00), Geometry (66.67 vs. 58.67), Number Theory (66.67 vs. 54.44), and Prealgebra (84.21 vs. 61.05), while not uniformly improving every category (e.g., Intermediate Algebra). When increasing the budget to 4%, SHIFT further improves to **64.67** on MATH-500, narrowing the gap to the full-data RLVR reference (66.00) while using only a small fraction of the pool. On AMC, SHIFT remains strong at 36.14 (still above the full-data reference 33.73), although the best OOD score at 4% is achieved by *Cluster* (40.96), highlighting that different heuristics may trade off in-domain performance and OOD transfer at larger budgets. Overall, these results show that SHIFT can reliably identify high-impact RLVR examples from an unlabeled pool and translate them into strong in-domain accuracy and robust low-budget generalization.

**Results on Medical Reasoning Tasks.** Table 2 reports RLVR results on MEDQA and three transfer benchmarks (MEDMCQA, PUBMEDQA, and MedXpertQA). Under the most stringent budget (0.1%), SHIFT is the strongest training-free selector overall: it achieves **50.35** on MEDQA (vs. 45.28 for *Random*), improves transfer to MEDMCQA (48.30) and PUBMEDQA (**74.20**), and yields particularly large gains on the clinically challenging MedXpertQA subsets, reaching **9.62** (R set) and **12.22** (U set) compared to 6.84 and 8.56 for *Random*. Notably, with only 0.1% data, SHIFT nearly matches the full-data RLVR performance on MedXpertQA-U (12.22 vs. 12.39), suggesting that selecting a small set of high-impact examples can be more effective

*Table 1.* **Mathematical reasoning under low budgets.** RLVR is trained on 2% and 4% subsets selected from MATH-500 and evaluated on MATH-500 (in-domain) and AMC (OOD). We report Pass@1 accuracy, with MATH-500 broken down by subject categories.

| Method | Train Size | Alg. | C. P. | Geo. | I. Alg. | N. T. | Prealg. | Precal. | MATH-500 | AMC |
|---|---|---|---|---|---|---|---|---|---|---|
| Qwen2.5-Math-1.5B | N/A | 55.56 | 33.33 | 20.00 | 27.27 | 44.44 | 57.89 | 29.41 | 40.00 | 27.71 |
| Qwen2.5-Math-1.5B | 100% | 88.89 | 58.33 | 60.00 | 39.39 | 77.78 | 78.95 | 52.94 | 66.00 | 33.73 |
| Random (avg) | 2% | 69.44 | 45.00 | 58.67 | 39.39 | 54.44 | 61.05 | 49.41 | 53.73 | 25.78 |
| Cluster | 2% | 61.11 | 41.67 | 40.00 | 39.39 | 38.89 | 31.58 | 47.06 | 44.67 | 25.30 |
| CoreSet | 2% | 66.67 | 25.00 | 26.67 | 36.36 | 50.00 | 52.63 | 52.94 | 47.33 | 25.30 |
| SC-Entropy | 2% | 75.00 | 41.67 | 60.00 | 39.39 | 44.44 | 68.42 | 52.94 | 56.00 | 31.33 |
| CoT Similarity | 2% | 75.00 | 58.33 | 53.33 | 39.39 | 66.67 | 63.16 | 52.94 | 58.67 | 37.35 |
| A-PPL | 2% | 72.22 | 41.67 | 53.33 | 30.30 | 61.11 | 73.68 | 52.94 | 55.33 | 31.33 |
| Q-PPL | 2% | 77.78 | 50.00 | 60.00 | 42.42 | 66.67 | 63.16 | 52.94 | 60.00 | 30.12 |
| SHIFT | 2% | 77.78 | 58.33 | 66.67 | 33.33 | 66.67 | 84.21 | 58.82 | 62.67 | 38.55 |
| Random (avg) | 4% | 76.67 | 46.67 | 57.33 | 40.61 | 58.89 | 70.53 | 47.06 | 54.53 | 31.08 |
| Cluster | 4% | 77.78 | 41.67 | 53.33 | 51.52 | 33.33 | 68.42 | 41.18 | 56.00 | 40.96 |
| CoreSet | 4% | 72.22 | 33.33 | 40.00 | 30.30 | 27.78 | 47.37 | 47.06 | 45.33 | 25.30 |
| SC-Entropy | 4% | 72.22 | 41.67 | 60.00 | 36.36 | 66.67 | 73.68 | 47.06 | 57.33 | 31.33 |
| CoT Similarity | 4% | 69.44 | 50.00 | 66.67 | 42.42 | 72.22 | 78.95 | 58.82 | 62.00 | 33.73 |
| A-PPL | 4% | 86.11 | 41.67 | 60.00 | 39.39 | 72.22 | 63.16 | 58.82 | 62.00 | 32.53 |
| Q-PPL | 4% | 75.00 | 50.00 | 60.00 | 45.45 | 44.44 | 73.68 | 47.06 | 58.00 | 31.33 |
| SHIFT | 4% | 80.56 | 50.00 | 60.00 | 51.52 | 66.67 | 78.95 | 52.94 | 64.67 | 36.14 |

*Table 2.* **Medical QA and transfer under low budgets.** RLVR is trained on 0.1% and 0.2% subsets selected from the MEDQA training pool and evaluated on MEDQA, MEDMCQA, PUBMEDQA, and MedXpertQA-R/U. Numbers are Pass@1 accuracy with greedy decoding.

| Method | Train Size | MedQA | MedMCQA | PubMedQA | MedXpertQA-R | MedXpertQA-U |
|---|---|---|---|---|---|---|
| Qwen3-1.7B | N/A | 32.91 | 40.98 | 66.20 | 2.36 | 4.92 |
| Qwen3-1.7B | 100% | 52.24 | 48.54 | 69.60 | 10.26 | 12.39 |
| Random (avg) | 0.1% | 45.28 | 45.91 | 70.56 | 6.84 | 8.56 |
| Cluster | 0.1% | 45.17 | 45.88 | 71.00 | 6.93 | 7.30 |
| CoreSet | 0.1% | 43.91 | 44.89 | 68.60 | 6.02 | 7.64 |
| SC-Entropy | 0.1% | 46.03 | 46.38 | 71.60 | 7.79 | 8.83 |
| CoT Similarity | 0.1% | 48.94 | 46.09 | 71.40 | 9.19 | 10.19 |
| Q-PPL | 0.1% | 43.99 | 45.53 | 69.60 | 5.80 | 8.49 |
| A-PPL | 0.1% | 40.77 | 45.17 | 69.00 | 4.51 | 7.30 |
| SHIFT | 0.1% | 50.35 | 48.30 | 74.20 | 9.62 | 12.22 |
| Random (avg) | 0.2% | 47.37 | 46.97 | 70.72 | 7.88 | 9.10 |
| Cluster | 0.2% | 49.65 | 47.30 | 69.00 | 9.35 | 11.54 |
| CoreSet | 0.2% | 46.43 | 46.48 | 70.40 | 7.15 | 8.66 |
| SC-Entropy | 0.2% | 47.05 | 47.37 | 72.40 | 7.25 | 9.17 |
| CoT Similarity | 0.2% | 49.18 | 45.85 | 71.00 | 9.03 | 9.68 |
| Q-PPL | 0.2% | 44.62 | 45.88 | 70.00 | 6.29 | 7.64 |
| A-PPL | 0.2% | 45.72 | 46.52 | 69.20 | 7.47 | 8.83 |
| SHIFT | 0.2% | 50.04 | 48.47 | 70.80 | 11.12 | 12.22 |

for clinical generalization than simply increasing the training pool. At 0.2%, SHIFT remains competitive on MEDQA (**50.04**) and achieves the best result on MedXpertQA-R (**11.12**), while other heuristics can be strong on specific datasets (e.g., *SC-Entropy* on PUBMEDQA). Overall, the pattern across budgets indicates that hidden-state-dynamics-driven selection is especially effective at acquiring examples that transfer to harder clinical reasoning settings, whereas diversity-only or surface-level difficulty/uncertainty heuristics exhibit less consistent transfer gains.

*Table 3.* **Ablation on medical QA.** CoreSet: farthest-first (FF) in question-embedding. FF-$\phi$: FF in coverage space $\phi$. QW-FF-$\phi$: quality-weighted FF using $\tilde{q}(x)$. $\phi_s = \mathbf{s}(x)$, $\phi_\Delta = \Delta(x)$, and $\phi_{s+\Delta} = [\mathbf{s}(x); \Delta(x)]$. Top: 0.1% budget; Bottom: 0.2%.

| Method | MedQA | MedMCQA | PubMedQA | MedXpertQA-R/U |
|---|---|---|---|---|
| Base | 32.91 | 40.98 | 66.20 | 2.36 / 4.92 |
| CoreSet | 43.91 | 44.89 | 68.60 | 6.02 / 7.64 |
| TopK-$\tilde{q}$ | 47.84 | 47.55 | 69.80 | 8.92 / 10.02 |
| FF-$\phi_{s+\Delta}$ | 47.53 | 47.37 | 73.00 | 8.49 / 9.85 |
| QW-FF-$\phi_s$ | 46.58 | 47.23 | 70.80 | 7.04 / 9.51 |
| QW-FF-$\phi_\Delta$ | 48.78 | 47.87 | 72.60 | 9.94 / 11.54 |
| QW-FF-$\phi_{s+\Delta}$ | 50.35 | 48.30 | 74.20 | 9.62 / 12.22 |
| CoreSet | 46.43 | 46.48 | 70.40 | 7.15 / 8.66 |
| TopK-$\tilde{q}$ | 49.25 | 46.52 | 71.20 | 8.17 / 9.00 |
| FF-$\phi_{s+\Delta}$ | 48.70 | 48.19 | 68.40 | 9.03 / 11.04 |
| QW-FF-$\phi_s$ | 46.03 | 47.69 | 69.60 | 7.47 / 7.64 |
| QW-FF-$\phi_\Delta$ | 49.25 | 47.73 | 72.20 | 9.19 / 11.38 |
| QW-FF-$\phi_{s+\Delta}$ | 50.04 | 48.47 | 70.80 | 11.12 / 12.22 |

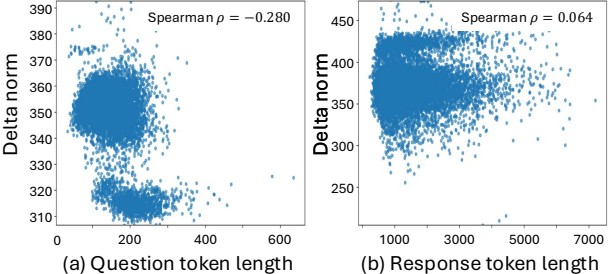

*Figure 2.* **RIRS is not a length proxy on MedQA.** RIRS magnitude $\|\Delta(x)\|_2$ versus (a) question token length and (b) response token length. Spearman correlations are provided.

### 4.3. Ablation Study

**Ablation on Medical QA.** Table 3 ablates the two ingredients of SHIFT, the coverage space $\phi$ for CoreSet selection and the quality-weighting by $\tilde{q}(x)$. Replacing the vanilla CoreSet baseline (FF in question-embedding space) with our RIRS-based coverage $\phi_{s+\Delta}$ consistently improves transfer, especially on PUBMEDQA and MedXpertQA-R/U (e.g., at 0.1%: 73.00 vs. 68.60 on PUBMEDQA). Adding quality-weighting further boosts performance: QW-FF-$\phi_\Delta$ strengthens clinical transfer (MedXpertQA-R/U reaches 9.94/11.54 at 0.1%), while QW-FF-$\phi_{s+\Delta}$ achieves the best overall trade-off, giving the top in-domain MEDQA accuracy at both budgets (50.35 at 0.1%, 50.04 at 0.2%) and the strongest MedXpertQA-R at 0.2% (11.12). Overall, $\phi_{s+\Delta}$ is key for robust coverage/transfer, and $\tilde{q}(x)$ provides complementary gains by prioritizing high-utility instances.

## 5. Analysis and Discussions

**RIRS vs. length.** Figure 2 shows that RIRS is essentially independent of response verbosity on MedQA: the correlation between $\|\Delta(x)\|_2$ and response length is negligible (Spearman $\rho = 0.064$). In contrast, $\|\Delta(x)\|_2$ exhibits a weak negative correlation with question length

*Table 4.* **RIRS rank vs. single-instance RLVR gain.** Top-5 and bottom-5 instances ranked by pre-RL RIRS; $\Delta$Pass@1 is measured on a held-out set relative to the base model (base Pass@1 = 40.0). Correlations are computed using the RIRS ranks, while scores are shown rounded for readability.

| RIRS Rank | 1 | 2 | 3 | 4 | 5 | 6 | 7 | 8 | 9 | 10 |
|---|---|---|---|---|---|---|---|---|---|---|
| RIRS Score | 5.61 | 5.61 | 5.61 | 5.60 | 5.60 | 4.61 | 4.60 | 4.59 | 4.57 | 4.49 |
| $\Delta$Pass@1 (%) | +17.67 | +13.91 | +12.58 | +15.58 | +14.00 | +7.50 | +11.08 | +12.00 | +9.58 | +2.25 |

($\rho = -0.280$), indicating that the hidden-state shift is not explained by longer inputs or longer generated traces. Overall, these trends support that RIRS captures reasoning dynamics beyond simple input/output length statistics.

**Direct validation of RIRS as a utility proxy.** To directly test whether pre-RL RIRS correlates with downstream RLVR gain, we conducted a pilot study on Qwen2.5-Math-1.5B / MATH-500. We selected 10 unlabeled instances (top-5 and bottom-5 by RIRS), ran single-instance RLVR on each, and measured per-instance $\Delta$Pass@1 on the same held-out validation set relative to the pre-RL base model (base Pass@1 = 40.0). As shown in Table 4, the pre-RL RIRS ranking exhibits a strong positive correlation with downstream gain: Spearman's $\rho = 0.818$ ($p = 0.0038$) and Kendall's $\tau = 0.644$ ($p = 0.0091$), where the correlations are computed using the RIRS ranks. This empirically supports our use of $\|\Delta(x)\|_2$ as a training-free utility surrogate.

**Generalization to larger and non-Qwen models.** To verify that SHIFT is not specific to small Qwen checkpoints, we extend the MATH-500 evaluation under the same 2% budget to two larger non-Qwen models: Llama-3-8B (Grattafiori et al., 2024) and Olmo-3-7B-Instruct (OLMo-3-7B) (Olmo et al., 2025). As shown in Table 5, SHIFT consistently outperforms Random sampling on both models and remains competitive with the strongest training-free baselines, suggesting that RIRS-driven selection transfers across model families and scales beyond 1.5B.

**Utility–coverage tradeoff across budgets.** SHIFT operates in a utility–coverage tradeoff regime. Under ultra-low budgets, selecting a few highly impactful samples is the critical factor for triggering initial adaptation, which is where the RIRS-based utility signal is most helpful. As the budget grows, the marginal benefit of prioritizing high-shift samples can diminish due to increasing redundancy among high-RIRS instances, and broader coverage becomes relatively more important, especially for OOD-style evaluations such as AMC. This may explain why clustering-based diversity methods occasionally surpass SHIFT at larger budgets.

**Selection-time compute.** SHIFT performs one greedy rollout ($T=0$) per candidate and only extracts anchor-level hidden states, so selection-time cost scales linearly with the

*Table 5.* **Cross-architecture results on MATH-500 (2% budget).**

| Model | Method | Pass@1 | Pass@8 |
|-------|--------|--------|--------|
| Llama-3-8B | Zero-shot | 20.67 | 54.67 |
|  | Full-Data (100%) | 27.83 | 59.33 |
|  | Random (avg) | 23.41 | 53.22 |
|  | SC-Entropy | 24.33 | 52.67 |
|  | CoT Similarity | 22.67 | 52.67 |
|  | Q-PPL | 24.50 | 54.67 |
|  | **SHIFT** | **25.17** | **57.33** |
| OLMo-3-7B | Zero-shot | 81.08 | 94.67 |
|  | Full-Data (100%) | 85.58 | 96.00 |
|  | Random (avg) | 81.67 | 94.00 |
|  | SC-Entropy | 82.91 | 94.00 |
|  | CoT Similarity | 81.25 | 94.67 |
|  | Q-PPL | 83.58 | 93.33 |
|  | **SHIFT** | **83.67** | **95.33** |

pool size $N$ and is dominated by a single decoding pass. In contrast, self-consistency selectors (e.g., SC-Entropy and CoT-Similarity) require $R$ stochastic rollouts per instance, yielding an $\approx R\times$ higher generation cost at selection time. Despite using only a single rollout, SHIFT outperforms these higher-compute baselines in Tables 1 and 2, indicating that its gains are not a byproduct of increased selection-time compute.

## 6. Conclusion

We studied *zero-RL-run* RLVR data selection: identifying a small set of high-impact training instances from a large unlabeled pool without labels, verifiable rewards, or training-time signals at selection time. We proposed SHIFT, a one-shot selector that uses a *single deterministic rollout* per candidate and extracts a *reasoning-induced representation shift* (RIRS) via the start-to-end hidden-state delta. SHIFTcombines an RIRS-based utility score with quality-weighted farthest-first CoreSet selection in an RIRS-augmented feature space, yielding compact subsets that scale to large pools. Across mathematical reasoning and medical QA under ultra-low budgets, SHIFT consistently outperforms training-free diversity and difficulty/uncertainty baselines, improves transfer to harder benchmarks, and remains compute-efficient at selection time (one rollout per instance). Ablations confirm that RIRS-based coverage and quality-weighting contribute complementary gains. Our current RIRS construction averages anchor hidden states across layers, which may dilute the most informative layers; exploring *layer selection* or learnable layer weighting could further strengthen the proxy. More broadly, extending RIRS beyond a single start–end delta (e.g., multi-anchor or token-level dynamics) and evaluating robustness across different model families, reward checkers, and reasoning domains

are promising directions.

## Limitations

Our work has several limitations. (i) *Theory–metric gap.* RIRS is a trajectory-level surrogate aggregated across layers, while our motivating theory is block-level; we therefore present the connection as motivation rather than derivation. (ii) *One-shot, static selection.* RIRS is computed from a single deterministic pre-RL rollout and does not adapt during training; degenerate or pathological rollouts (e.g., truncated or repetitive traces) may produce noisy estimates. (iii) *Scope of validation.* While we report results on 1.5B–8B models across Qwen, Llama, and OLMo families, robustness across broader model families, reward checkers, and reasoning domains remains to be established. (iv) *Safety-sensitive domains.* For medical QA, final accuracy alone is insufficient; harmful vs. harmless error stratification is an important future evaluation axis.

## Impact Statement

This paper presents work whose goal is to advance the field of Machine Learning, specifically the efficient adaptation of large language models for reasoning tasks. By substantially reducing the annotation and training cost of RLVR, SHIFT may broaden access to reasoning-capable LLMs in low-resource and specialized domains such as medical QA. We acknowledge two potential risks. First, any training-free selector, including SHIFT, can systematically prefer certain instance types and thus inherit or amplify biases present in the unlabeled pool; this is particularly consequential in safety-sensitive domains. Second, lowering the cost of adapting reasoning models may reduce barriers to misuse. We encourage users to combine SHIFT with downstream safety evaluation, bias auditing, and domain-appropriate review before deployment.

## Acknowledgements

This research was supported by The Commonwealth of Australia under the Medical Research Future Fund No. NCRI000074. This research was undertaken with the assistance of resources from the National Computational Infrastructure (NCI Australia), an NCRIS enabled capability supported by the Australian Government. This work was also supported by Monash eResearch capabilities, including M3.

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

# A. Verifiable reward and GRPO objective

**From selection to labeled RLVR.** After selecting a compact subset $S \subset \mathcal{U}$ with $|S| = B$, we obtain ground-truth annotations for $S$ and form a labeled set $\mathcal{D}_S = \{(x, a^\star)\}$, where $a^\star$ denotes the verified target answer (e.g., the correct option index in multiple-choice QA). We then perform on-policy reinforcement learning with a *verifiable* reward computed against $a^\star$.

**Verifiable reward.** For each input $x$, we sample a group of $N$ responses $\{y_i\}_{i=1}^{N} \sim \pi_{\theta_{\text{old}}}(\cdot|x)$ and compute a reward

$$r_i \ = \ r(y_i, a^\star) \in [0, 1], \tag{12}$$

where $r(\cdot, \cdot)$ is a task-specific verification function (e.g., exact-match between the extracted final answer from $y_i$ and $a^\star$). This provides a reliable supervision signal without relying on self-consistency heuristics.

**GRPO with group-relative advantages.** Within the group of $N$ samples for the same $x$, we compute the standardized group-relative advantage

$$\hat{A}^i \ = \ \frac{r_i - \text{mean}(\{r_j\}_{j=1}^N)}{\text{std}(\{r_j\}_{j=1}^N) + \epsilon}, \qquad i = 1, \ldots, N. \tag{13}$$

Let $\rho_t^{(i)}(\theta)$ be the token-level importance ratio

$$\rho_t^{(i)}(\theta) \ = \ \frac{\pi_\theta\left(a_t^{(i)} \mid s_t^{(i)}\right)}{\pi_{\theta_{\text{old}}}\left(a_t^{(i)} \mid s_t^{(i)}\right)}. \tag{14}$$

We optimize the clipped surrogate objective

$$\ell_{\text{GRPO}} = \min\left[\rho_t^{(i)} \hat{A}^i, \ \text{clip}(\rho_t^{(i)}, 1-\epsilon_c, 1+\epsilon_c) \hat{A}^i\right], \tag{15}$$

where $\epsilon_c$ is the clipping parameter.

**KL regularization.** To prevent excessive policy drift, we add a KL penalty to a fixed reference policy $\pi_{\text{ref}}$ (set to the pre-adaptation base model $\pi_{\theta_0}$):

$$\ell_{\text{KL}} = \frac{\mathbb{E}_{(i,t)\in\mathcal{B}}\left[m_t^{(i)} D_{\text{KL}}\left(\pi_\theta(\cdot \mid s_t^{(i)}) \,\|\, \pi_{\text{ref}}(\cdot \mid s_t^{(i)})\right)\right]}{\mathbb{E}_{(i,t)\in\mathcal{B}}\left[m_t^{(i)}\right] + \epsilon}. \tag{16}$$

Here $m_t^{(i)} \in \{0, 1\}$ is an optional mask; by default we set $m_t^{(i)} = 1$ for all tokens (i.e., standard KL). Our final objective is

$$\max_\theta \ \mathbb{E}_{(x,a^\star)\sim\mathcal{D}_S}\left[\mathbb{E}_{\{y_i\}_{i=1}^N \sim \pi_{\theta_{\text{old}}}}\left[\frac{1}{N}\sum_{i=1}^N \sum_t \ell_{\text{GRPO}}\right] - \beta\,\ell_{\text{KL}}\right], \tag{17}$$

where $\beta$ controls the KL strength.

# B. Evaluation Normalization

### B.1. Prompt Templates

For completeness, we report the fixed prompt templates used in all experiments. A single template is used for open-ended reasoning tasks (MATH-500), and a single template is used for multiple-choice benchmarks (Medical QA). All methods and baselines share exactly the same prompts.

### B.2. Answer Extraction and Reward Computation

We evaluate math and QA benchmarks using the public `mathruler` grader, which standardizes answer extraction and normalization across datasets.[1]

---

[1] We use the official implementation from https://github.com/bytedance/mathruler.

---

**Prompt Templates**

**Open-Ended Reasoning Tasks.**
*"You FIRST think about the reasoning process as an internal monologue and then provide the final answer.*
*The reasoning process MUST BE enclosed within* `<think> </think>` *tags.*
*The final answer MUST BE put in* $\backslash boxed\{\}$*."*

**Multiple-Choice QA Tasks.**
*"You should FIRST think about the reasoning process as an internal monologue and then provide the final answer.*
*The reasoning process MUST BE enclosed within* `<think> </think>` *tags.*
*The final answer MUST BE a single choice letter (A, B, C, etc.) and MUST be put in* $\backslash boxed\{X\}$*, where X is the selected letter."*

*Table 6.* Prompt templates used for reasoning with CoT.

**Answer extraction and normalization.** For all tasks, the model is instructed to place its final prediction inside a $\backslash boxed\{\}$ macro. At evaluation time, we extract the content of the last $\backslash boxed\{\}$ using the `extract_boxed_content` helper, and pass the resulting string to `grade_answer(answer, ground_truth)`, which applies the standard Math-Ruler normalization pipeline (whitespace and case normalization, LaTeX simplification, and numeric equivalence checks) and returns a binary correctness flag.

## C. Additional Training Details

All training settings are shared across datasets and baselines. We follow the default GRPO configuration of the EasyR1 framework[2].

**Data and rollout configuration.** Inputs are truncated to the maximum prompt and response lengths specified in the main paper. During training, we use a fixed rollout strategy with $n = 8$ trajectories per input and temperature-based decoding (temperature 0.7, top-$p = 0.95$), together with a global rollout batch size of 12. At evaluation time, we switch to deterministic decoding ($n = 1$, temperature 0.0, top-$p = 1.0$).

**Optimization and schedule.** We use GRPO with KL regularization and optimize the policy using AdamW with weight decay and gradient clipping following the EasyR1 defaults. Padding-free training, dynamic batching, gradient checkpointing, and fully sharded data parallelism (FSDP) with parameter and optimizer offloading are enabled in all runs. Unless otherwise noted, all models share the same core hyperparameters: a learning rate of $2.0 \times 10^{-6}$, weight decay of $1.0 \times 10^{-2}$, and a maximum gradient norm of 1.0.

---

[2]https://github.com/hiyouga/EasyR1

