# A. Per-seed Results for Random Sampling

**0.1% budget (upper block).** Random subset training yields consistent gains over the base model across all benchmarks. Averaged over five seeds, performance improves from 32.91 to 45.28 on MEDQA (+12.37), from 40.98 to 45.91 on MEDMCQA (+4.93), and from 66.20 to 70.56 on PUBMEDQA (+4.36). MEDXPERTQA also benefits notably (R: 2.36→6.84, +4.48; U: 4.92→8.56, +3.64). Across seeds, the variability is modest (e.g., MEDQA ranges 43.68–46.11; PUBMEDQA ranges 69.40–71.80), indicating stable improvements under random sampling.

**0.2% budget (lower block).** Increasing the budget further improves the average performance on most datasets, with MEDQA rising to 47.37 (+14.46 over base) and MEDMCQA to 46.97 (+5.99). MEDXPERTQA shows the largest absolute gains (R: 2.36→7.88, +5.52; U: 4.92→9.10, +4.18), although the seed sensitivity becomes more pronounced for MEDXPERTQA-R (6.45–9.13) and MEDXPERTQA-U (8.49–10.53). Overall, the 0.2% setting provides a stronger mean performance while exhibiting slightly higher variance on the more challenging MEDXPERTQA metrics.

*Table 4.* **Per-seed Random baselines on mathematical reasoning.** We run Random sampling with 5 seeds for each budget (2% and 4% of the MATH-500 selection pool). We report Pass@1 on MATH-500 (with category breakdown) and AMC (OOD).

| Method | Train Size | Alg. | C. P. | Geo. | I. Alg. | N. T. | Prealg. | Precal. | MATH500 | AMC |
|---|---|---|---|---|---|---|---|---|---|---|
| Qwen2.5-Math-1.5B | 0% | 55.56 | 33.33 | 20.00 | 27.27 | 44.44 | 57.89 | 29.41 | 40.00 | 27.71 |
| Random 1 | 2% | 69.44 | 75.00 | 60.00 | 45.45 | 72.22 | 68.42 | 52.94 | 57.33 | 27.71 |
| Random 2 | 2% | 58.33 | 58.33 | 60.00 | 42.42 | 50.00 | 63.16 | 41.18 | 52.67 | 25.30 |
| Random 3 | 2% | 69.44 | 33.33 | 60.00 | 30.30 | 44.44 | 57.89 | 52.94 | 50.67 | 21.69 |
| Random 4 | 2% | 75.00 | 41.67 | 53.33 | 39.39 | 38.89 | 52.63 | 35.29 | 50.67 | 22.89 |
| Random 5 | 2% | 75.00 | 16.67 | 60.00 | 39.39 | 66.67 | 63.16 | 64.71 | 57.33 | 31.33 |
| Random (avg) | 2% | 69.44 | 45.00 | 58.67 | 39.39 | 54.44 | 61.05 | 49.41 | 53.73 | 25.78 |
| Random 1 | 4% | 80.56 | 58.33 | 53.33 | 51.52 | 77.78 | 78.95 | 52.94 | 48.00 | 34.94 |
| Random 2 | 4% | 69.44 | 50.00 | 60.00 | 42.42 | 44.44 | 63.16 | 47.06 | 54.67 | 32.53 |
| Random 3 | 4% | 77.78 | 41.67 | 66.67 | 42.42 | 50.00 | 68.42 | 41.18 | 57.33 | 27.71 |
| Random 4 | 4% | 83.33 | 33.33 | 46.67 | 30.30 | 61.11 | 73.68 | 41.18 | 55.33 | 25.30 |
| Random 5 | 4% | 72.22 | 50.00 | 60.00 | 36.36 | 61.11 | 68.42 | 52.94 | 57.33 | 34.94 |
| Random (avg) | 4% | 76.67 | 46.67 | 57.33 | 40.61 | 58.89 | 70.53 | 47.06 | 54.53 | 31.08 |

*Table 5.* **Per-seed Random baselines on medical QA.** Models are trained on subsets selected from the MEDQA training pool (0.1% and 0.2% budgets), and evaluated on MEDQA as well as three benchmarks: MEDMCQA, PUBMEDQA, and MedXpertQA (u/r).

| Method | Train Size | MedQA | MedmcQA | Pubmedqa | Medxpertqa-r | Medxpertqa-u |
|---|---|---|---|---|---|---|
| Qwen3-1.7B | 0% | 32.91 | 40.98 | 66.20 | 2.36 | 4.92 |
| Random 1 | 0.1% | 45.64 | 45.74 | 69.40 | 6.82 | 8.66 |
| Random 2 | 0.1% | 45.48 | 45.88 | 71.80 | 6.93 | 8.32 |
| Random 3 | 0.1% | 46.11 | 46.66 | 70.20 | 7.42 | 8.15 |
| Random 4 | 0.1% | 43.68 | 45.77 | 70.80 | 6.23 | 8.49 |
| Random 5 | 0.1% | 45.48 | 45.49 | 70.60 | 6.82 | 9.17 |
| Random (avg) | 0.1% | 45.28 | 45.91 | 70.56 | 6.84 | 8.56 |
| Random 1 | 0.2% | 48.55 | 47.05 | 71.40 | 9.13 | 10.53 |
| Random 2 | 0.2% | 46.74 | 47.09 | 71.20 | 8.44 | 8.83 |
| Random 3 | 0.2% | 47.76 | 46.38 | 70.80 | 8.28 | 8.66 |
| Random 4 | 0.2% | 46.90 | 47.59 | 68.80 | 6.45 | 8.49 |
| Random 5 | 0.2% | 46.90 | 46.73 | 71.40 | 7.09 | 9.00 |
| Random (avg) | 0.2% | 47.37 | 46.97 | 70.72 | 7.88 | 9.10 |

### A.1. Verifiable reward and GRPO objective

**From selection to labeled RLVR.** After selecting a compact subset $S \subset \mathcal{U}$ with $|S| = B$, we obtain ground-truth annotations for $S$ and form a labeled set $\mathcal{D}_S = \{(x, a^\star)\}$, where $a^\star$ denotes the verified target answer (e.g., the correct option index in multiple-choice QA). We then perform on-policy reinforcement learning with a *verifiable* reward computed against $a^\star$.

**Verifiable reward.** For each input $x$, we sample a group of $N$ responses $\{y_i\}_{i=1}^N \sim \pi_{\theta_{\text{old}}}(\cdot|x)$ and compute a reward

$$r_i \;=\; r(y_i, a^\star) \in [0, 1], \tag{12}$$

where $r(\cdot, \cdot)$ is a task-specific verification function (e.g., exact-match between the extracted final answer from $y_i$ and $a^\star$). This provides a reliable supervision signal without relying on self-consistency heuristics.

**GRPO with group-relative advantages.** Within the group of $N$ samples for the same $x$, we compute the standardized group-relative advantage

$$\hat{A}^i \;=\; \frac{r_i - \text{mean}(\{r_j\}_{j=1}^N)}{\text{std}(\{r_j\}_{j=1}^N) + \epsilon}, \qquad i = 1, \dots, N. \tag{13}$$

Let $\rho_t^{(i)}(\theta)$ be the token-level importance ratio

$$\rho_t^{(i)}(\theta) \;=\; \frac{\pi_\theta\Big(a_t^{(i)} \mid s_t^{(i)}\Big)}{\pi_{\theta_{\text{old}}}\Big(a_t^{(i)} \mid s_t^{(i)}\Big)}. \tag{14}$$

We optimize the clipped surrogate objective

$$\ell_{\text{GRPO}} = \min\Big[\rho_t^{(i)} \hat{A}^i, \; \text{clip}(\rho_t^{(i)}, 1-\epsilon_c, 1+\epsilon_c) \hat{A}^i\Big], \tag{15}$$

where $\epsilon_c$ is the clipping parameter.

**KL regularization.** To prevent excessive policy drift, we add a KL penalty to a fixed reference policy $\pi_{\text{ref}}$ (set to the pre-adaptation base model $\pi_{\theta_0}$):

$$\ell_{\text{KL}} = \frac{\mathbb{E}_{(i,t)\in\mathcal{B}}\Big[m_t^{(i)} D_{\text{KL}}\Big(\pi_\theta(\cdot \mid s_t^{(i)}) \,\|\, \pi_{\text{ref}}(\cdot \mid s_t^{(i)})\Big)\Big]}{\mathbb{E}_{(i,t)\in\mathcal{B}}\big[m_t^{(i)}\big] + \epsilon}. \tag{16}$$

Here $m_t^{(i)} \in \{0, 1\}$ is an optional mask; by default we set $m_t^{(i)} = 1$ for all tokens (i.e., standard KL). Our final objective is

$$\max_\theta \; \mathbb{E}_{(x,a^\star)\sim\mathcal{D}_S}\left[\mathbb{E}_{\{y_i\}_{i=1}^N \sim \pi_{\theta_{\text{old}}}}\left[\frac{1}{N}\sum_{i=1}^N \sum_t \ell_{\text{GRPO}}\right] - \beta\,\ell_{\text{KL}}\right], \tag{17}$$

where $\beta$ controls the KL strength.

## B. Evaluation Normalization

### B.1. Prompt Templates

For completeness, we report the fixed prompt templates used in all experiments. A single template is used for open-ended reasoning tasks (MATH500), and a single template is used for multiple-choice benchmarks (Medical QA). All methods and baselines share exactly the same prompts.

### B.2. Answer Extraction and Reward Computation

We evaluate math and QA benchmarks using the public `mathruler` grader, which standardizes answer extraction and normalization across datasets.[1]

---

[1] We use the official implementation from https://github.com/bytedance/mathruler.

---

**Prompt Templates**

**Open-Ended Reasoning Tasks.**
*"You FIRST think about the reasoning process as an internal monologue and then provide the final answer.*
*The reasoning process MUST BE enclosed within `<think>` `</think>` tags.*
*The final answer MUST BE put in* $\backslash boxed\{\}$."

**Multiple-Choice QA Tasks.**
*"You should FIRST think about the reasoning process as an internal monologue and then provide the final answer.*
*The reasoning process MUST BE enclosed within `<think>` `</think>` tags.*
*The final answer MUST BE a single choice letter (A, B, C, etc.) and MUST be put in* $\backslash boxed\{X\}$*, where X is the selected letter."*

*Table 6.* Prompt template used for reasoning with CoT.

**Answer extraction and normalization.** For all tasks, the model is instructed to place its final prediction inside a $\backslash boxed\{\}$ macro. At evaluation time, we extract the content of the last $\backslash boxed\{\}$ using the `extract_boxed_content` helper, and pass the resulting string to `grade_answer(answer, ground_truth)`, which applies the standard Math-Ruler normalization pipeline (whitespace and case normalization, LaTeX simplification, and numeric equivalence checks) and returns a binary correctness flag.

## C. Additional Training Details

All training settings are shared across datasets and baselines. We follow the default GRPO configuration of the Easy-R1 framework[2] and only modify the entropy-band and forking-token components described in the main text.

**Data and rollout configuration.** Our training data is constructed from the same pool of math and QA benchmarks used in evaluation. Inputs are truncated to the maximum prompt and response lengths specified in the main paper. During training, we use a fixed rollout strategy with $n = 8$ trajectories per input and temperature-based decoding (temperature 0.7, top-$p = 0.95$), together with a global rollout batch size of 12. At evaluation time, we switch to deterministic decoding ($n = 1$, temperature 0.0, top-$p = 1.0$).

**Optimization and schedule.** We use GRPO with KL regularization and optimize the policy using AdamW with weight decay and gradient clipping following the Easy-R1 defaults. Padding-free training, dynamic batching, gradient checkpointing, and fully sharded data parallelism (FSDP) with parameter and optimizer offloading are enabled in all runs. Unless otherwise noted, all models share the same core hyperparameters: a learning rate of $2.0 \times 10^{-6}$, weight decay of $1.0 \times 10^{-2}$, and a maximum gradient norm of 1.0.

---

[2] https://github.com/hiyouga/EasyR1