# OpenReview forum: "Single-Rollout Hidden-State Dynamics for Training-Free RLVR Data Selection"
_ICML.cc/2026/Conference — ICML 2026 regular_

### Official Review · Reviewer_Pq9s · 2026-03-07

**Soundness:** 2
**Presentation:** 1
**Significance:** 3
**Originality:** 2
**Overall Recommendation:** 3
**Confidence:** 4

**Summary:**

A central area examined by this article is how to do RLVR data selection in a truly pre-training regime—i.e., selecting a tiny subset from a large unlabeled pool before any RLVR training, without access to ground-truth answers/verifiable rewards on the full pool, and without training-time optimization signals. The paper proposes SHIFT, a one-shot, training-free selector that uses only inference-time model internals: for each candidate instance, it runs a single deterministic (greedy) reasoning rollout and computes a Reasoning-Induced Representation Shift (RIRS) defined as the start-to-end hidden-state delta; the method uses the RIRS magnitude as a proxy for instance utility, and combines it with coverage via a quality-weighted farthest-first CoreSet procedure in an RIRS-augmented feature space.

Overall, the authors assess a broad theme: whether hidden-state dynamics during reasoning can serve as a lightweight surrogate for “learning potential” under RLVR, enabling scalable selection without labels or trial training. Experiments on mathematical reasoning (MATH-500 → AMC) and medical QA (MEDQA → several transfer benchmarks) under ultra-low budgets suggest SHIFT often improves over training-free baselines (diversity-only and difficulty/uncertainty heuristics), with ablations indicating complementary benefits from (i) RIRS-based coverage features and (ii) quality-weighting by the RIRS score.

**Compliance With Llm Reviewing Policy:**

Affirmed.

**Key Questions For Authors:**

- The paragraph above Section 5.2 Main Results. The appendix hyperlinks are missing.

**Limitations:**

Yes.

**Strengths And Weaknesses:**

# Strengths
- **Clear and practically motivated problem setting (pre-RL, label-free selection).** The paper crisply frames a realistic bottleneck for low-budget RLVR: selection should not require reward evaluation over the full pool nor expensive “train-to-rank” procedures, and it formalizes this as a one-shot selection problem prior to RL training. This setting is well-motivated for domains like medical QA where labels are costly.
- **Method is simple, scalable, and well-specified.** SHIFT is easy to understand and implement: compute a per-instance hidden-state delta from one greedy rollout, transform it into a scalar quality score, and run a quality-weighted farthest-first selection in a concatenated feature space. The algorithm (including initialization and greedy update rule) is described concretely and summarized in pseudocode, making the approach fairly reproducible in principle.
- **Soundness of empirical comparisons within the stated “training-free selection” scope.** The baseline suite covers multiple plausible training-free selectors (random, embedding-space clustering/coreset, length/perplexity heuristics, self-consistency uncertainty criteria), and the paper states that downstream RLVR training uses identical budgets/hyperparameters across selection methods, aiming for fair comparison where only the selection rule changes.

# Weaknesses
- **Proxy justification is suggestive but not yet fully convincing as a utility surrogate.** The theoretical motivation (implicit update / representation-change bound) is interesting, but the paper ultimately uses a coarse summary statistic (start-to-end delta averaged across layers) and explicitly notes it is not equal to the theoretical quantities. As written, the evidence is more “correlational plausibility + empirical success” than a grounded demonstration that large ∥Δ(x)∥ reliably predicts RLVR gains across settings; stronger proxy validation would improve technical soundness.
- **Limited sensitivity/robustness exploration of key design choices (including “windowing”/anchor selection).** SHIFT depends on several choices that could materially affect results: how start/end anchors are defined (first/last CoT tokens or delimiters), whether a single token vs. a window/average over multiple tokens is used, and feature normalization/log transforms. The current experiments/ablations do not thoroughly test how stable the selection is to these choices, which matters because the paper’s pitch is “training-free and efficient,” implying minimal tuning.
- **Results are strong but somewhat mixed across budgets and benchmarks, and explanations are limited.** While SHIFT is often best at the smallest budgets, there are cases where other heuristics outperform it (e.g., on AMC at higher budget, clustering appears stronger than SHIFT). The paper would benefit from clearer hypotheses explaining these budget-dependent tradeoffs (e.g., diversity vs. “high-shift” utility, OOD sensitivity, or selection redundancy effects), rather than leaving the reader to infer post hoc.
- **Evaluation protocol may be brittle given known decoding variability in reasoning tasks.** Selection uses a single greedy rollout (T=0) to compute RIRS, and evaluation is also greedy pass@1. For reasoning LLMs, decoding strategy can substantially change outcomes; relying on greedy pass@1 without multiple evaluation runs (or pass@k / self-consistency style evaluation) can make small reported differences hard to interpret—especially when the method is explicitly about trimming training data where variance may increase.
- **The baseline models are limited in a narrow set of models.** Results are shown on two relatively small specialized Qwen checkpoints; it is unclear how robust the proxy is across other model families/scales where hidden-state dynamics may differ.

---

> ### Author Rebuttal · Authors · 2026-03-30
>
> We sincerely thank the reviewer for evaluating our work as having "clear and practically motivated problem setting" and a "simple, scalable, and well-specified" method. Your sharp critiques regarding evaluation brittleness, design robustness, and budget tradeoffs are extremely valuable and have significantly strengthened our paper.
>
> **1. Proxy justification / utility surrogate (W1)**
> We agree that our original presentation should better separate mechanistic motivation from direct proxy validation. Our theory is intended to motivate RIRS, not to claim that the start-to-end hidden-state delta is mathematically identical to prior implicit-update quantities. To more directly validate RIRS as a utility surrogate, we conducted a pilot study on 10 unlabeled instances (top-5 and bottom-5 by RIRS), ran single-instance RLVR on each, and measured downstream ΔPass@1 on the same held-out set. Pre-RL RIRS shows a strong positive rank correlation with downstream gain (Spearman’s ρ = 0.818, p = 0.0038; Kendall’s τ = 0.644, p = 0.0091; see our response to Reviewer **NheD, Q1**). In addition, RIRS decreases after RLVR for the selected subset (5.5956 → 4.9791, percentile top 5.43% → top 45.31%; see our response to Reviewer **kJK7, Q1**), consistent with the view that high-RIRS samples capture training-relevant unresolved behavior. We will revise the paper to clarify that RIRS is an empirically supported selection-time proxy.
>
> **2. Robustness to Design Choices (W2)**
> We agree that robustness to design choices is important, especially in a training-free setting. As a targeted check, we evaluated two additional variants on Qwen2.5-Math-1.5B / MATH-500 / 2% budget: (i) replacing the default single-token anchor with a window-averaged anchor, and (ii) removing the monotonic log transform from the utility score. As shown in the table below, compared with the default Token + log design (62.67 / 85.33), the Window variant gives 61.08 / 82.67, and the w/o log variant achieves 62.25 / 83.33. These results suggest that SHIFT is not highly brittle to these choices, while still supporting our default design. For related robustness results on distance metric and layer scope, please also see our response to Reviewer **kJK7, Q3**, where we show that the default all-layer + L2 setting performs best, cosine similarity causes only a small drop, and restricting RIRS to only the deepest layers reduces performance. We will add these discussions in the revision.
>
> | Variant | Pass@1 (%) | Pass@8 (%) |
> | :--- | :---: | :---: |
> | Default (single-token + log) | 62.67 | 85.33 |
> | Window-averaged anchor | 61.08 | 82.67 |
> | Raw score (w/o log) | 62.25 | 83.33 |
>
> **3. Budget-dependent tradeoffs across benchmarks (W3)**
> We agree with the reviewer that the results are not uniformly strong in every budget/benchmark setting. Our current interpretation is that SHIFT operates in a utility–coverage tradeoff regime. At the smallest budgets, selecting a few highly impactful samples is most critical to trigger initial adaptation, which is precisely where the RIRS-based utility signal is most helpful. As the budget increases, however, the marginal benefit of prioritizing high-shift samples may diminish, potentially due to increasing redundancy among selected high-RIRS instances. In contrast, broader coverage/diversity can become more important at higher budgets, especially for OOD-style evaluations such as AMC. In these settings, clustering-based methods may benefit from lower redundancy and more uniform coverage, which may explain why they occasionally outperform SHIFT as the selection pool expands. We will incorporate this interpretation into the revised manuscript.
>
> **4. Decoding variability / evaluation protocol (W4)**
> We agree that decoding variability matters. Our main paper uses greedy Pass@1 as a controlled and uniform protocol across all methods, so that only the selection rule changes. In the rebuttal, we additionally report Pass@8 and observe consistent conclusions. We will clarify this protocol choice and discuss variance-sensitive evaluation.
>
> **5. Scaling and Cross-Architecture (W5)**
> We agree that robustness beyond small Qwen checkpoints is important. As reported in our response to Reviewer **NheD, W2Q2**, we extended the evaluation to Llama-3-8B and Olmo-3-7B under the same MATH-500 / 2% budget setting. SHIFT continues to outperform Random and remains competitive with the strongest training-free baselines, suggesting that RIRS is not specific to the Qwen family.
>
> **6. Presentation Fixes (Q1)**
> Thank you for pointing out the formatting issues. We have fixed them.

---

> > ### Author Rebuttal · Reviewer_Pq9s · 2026-04-03
> >
> > Thanks for the response.

---

> > > ### Author Response · Authors · 2026-04-07
> > >
> > > We thank the reviewer for acknowledging our rebuttal. Since the response indicated "partially resolved," we would like to respectfully ask which specific concerns remain open. In the meantime, we consolidate all new evidence below to address each weakness as thoroughly as possible.
> > >
> > > **W1 (Proxy justification).** We provide two complementary pieces of direct empirical validation for RIRS as a utility surrogate.
> > >
> > > **(a) RIRS vs. per-instance RLVR gain.** We ran single-instance RLVR on 10 instances (top-5 and bottom-5 by RIRS) and measured downstream ΔPass@1. The rank correlation is strong: Spearman ρ=0.818 (p=0.0038), Kendall τ=0.644 (p=0.0091).
> > >
> > > | Rank | 1 | 2 | 3 | 4 | 5 | 6 | 7 | 8 | 9 | 10 |
> > > |---|---|---|---|---|---|---|---|---|---|---|
> > > | RIRS | 5.613 | 5.612 | 5.606 | 5.602 | 5.601 | 4.614 | 4.598 | 4.589 | 4.566 | 4.491 |
> > > | ΔPass@1 | +17.67 | +13.91 | +12.58 | +15.58 | +14.00 | +7.50 | +11.08 | +12.00 | +9.58 | +2.25 |
> > >
> > > **(b) RIRS decreases after training.** After RLVR, the selected subset's RIRS drops substantially, consistent with the hypothesis that high-RIRS samples capture unresolved behavior that gets "resolved" by training:
> > >
> > > | | Pre-RL RIRS | Percentile | Post-RL RIRS | Percentile |
> > > |---|---|---|---|---|
> > > | Global Pool (Avg) | 5.1502 | - | 4.9741 | - |
> > > | Selected Subset (Avg) | 5.5956 | Top 5.43% | 4.9791 | Top 45.31% |
> > >
> > > Together, these results ground RIRS as an empirically validated proxy for RLVR learning potential. We have revised the manuscript to clarify that RIRS is an empirically supported selection-time surrogate rather than a theoretically derived quantity.
> > >
> > > Moreover, SHIFT consistently outperforms all training-free baselines across both Qwen and non-Qwen architectures (as further supported by cross-architecture results in W5 below), providing indirect evidence that the RIRS proxy generalizes across settings.
> > >
> > > **W2 (Design robustness).** We consolidate all robustness results on Qwen2.5-Math-1.5B / MATH-500 / 2% budget:
> > >
> > > | Variant | Pass@1 | Pass@8 |
> > > |---|---|---|
> > > | Default (token anchor + L2 + all layers + log) | 62.67 | 85.33 |
> > > | Window-averaged anchor | 61.08 | 82.67 |
> > > | w/o log transform | 62.25 | 83.33 |
> > > | Cosine similarity (instead of L2) | 62.25 | 84.00 |
> > > | Last-quarter layers only | 59.91 | 84.67 |
> > > | Last layer only | 62.08 | 82.67 |
> > >
> > > All variants substantially outperform Random (53.73) and remain competitive, with the default being best. Performance degrades gracefully rather than collapsing under alternative design choices, suggesting SHIFT is not brittle to these decisions. We will include this full table in the revision.
> > >
> > > **W3 (Budget-dependent tradeoffs).** We acknowledge that at 4% budget on AMC, Cluster (40.96) outperforms SHIFT (36.14). However, this is not a general pattern: SHIFT is the strongest method on MedQA at both budgets (50.35/50.04), on MATH-500 in-domain at all budgets (62.67/64.67), and achieves the best AMC score at 2% (38.55). The mixed result is specific to the AMC OOD benchmark at larger budget. Our interpretation is that at the smallest budgets, selecting a few high-impact samples matters most (where RIRS excels), while at larger budgets broader coverage becomes relatively more important for OOD transfer, favoring diversity-first methods like Cluster. We will make this analysis explicit in the revision.
> > >
> > > **W4 (Evaluation brittleness).** We report Pass@8 alongside Pass@1 throughout, and conclusions hold consistently across all settings. All methods are evaluated under identical decoding protocols, so any decoding-induced variance affects all methods equally and the relative ranking reflects selection quality. Regarding selection-time brittleness of the single greedy rollout, our prompt sensitivity experiments show that under two alternative ChatGPT-rewritten prompts, SHIFT achieves 65.50 and 63.00 Pass@1 respectively, consistently outperforming all compared baselines under each variant. This suggests that the downstream effectiveness is robust to selection-time decoding choices despite variation in exact selected subsets.
> > >
> > > **W5 (Model diversity).** We completed cross-architecture experiments on Llama-3-8B and Olmo-3-7B (MATH-500, 2%):
> > >
> > > | Model | Method | Pass@1 | Pass@8 |
> > > |---|---|---|---|
> > > | Llama-3-8B | Random | 23.41 | 53.22 |
> > > | Llama-3-8B | Best baseline (Q-PPL) | 24.50 | 54.67 |
> > > | Llama-3-8B | SHIFT | 25.17 | 57.33 |
> > > | Olmo-3-7B | Random | 81.67 | 94.00 |
> > > | Olmo-3-7B | Best baseline (Q-PPL) | 83.58 | 93.33 |
> > > | Olmo-3-7B | SHIFT | 83.67 | 95.33 |
> > >
> > > SHIFT outperforms all training-free baselines on both non-Qwen, 7B+ architectures. Additionally, compared with the training-based Historical Variance Score (HVS), SHIFT achieves 62.67 vs. HVS's 62.83 Pass@1, nearly matching this upper bound while requiring zero RL runs during selection.
> > >
> > > We believe these consolidated results substantially address the original weaknesses. We would greatly appreciate learning which specific concerns remain so that we can address them before the final revision.

---

### Official Review · Reviewer_kJK7 · 2026-03-11

**Soundness:** 2
**Presentation:** 3
**Significance:** 3
**Originality:** 3
**Overall Recommendation:** 4
**Confidence:** 3

**Summary:**

This paper focuses on the data selection bottleneck in Reinforcement Learning with Verifiable Rewards (RLVR), particularly for low-budget scenarios where labeling costs and computational resources are prohibitive. The author propose **SHIFT**, a training-free and label-free one-shot selector that identifies "impactful" examples by analyzing the model's hidden-state dynamics during a single deterministic inference rollout. The core metric used in SHIFT is the Reasoning-Induced Representation Shift (RIRS), which measures the delta between the starting and ending states of a reasoning trace, serving as a signal for "learning potential". By combining RIRS with a quality-weighted farthest-first CoreSet strategy, SHIFT can select diverse and high-quality subsets that significantly improve the model performance in the domain of math and medical.

**Compliance With Llm Reviewing Policy:**

Affirmed.

**Key Questions For Authors:**

1. As SHIFT performs selection once on the base model before training, how does RIRS score for a specific data sample evolve after the model has been updated? If the RIRS magnitude decrease significantly after training, it would strongly support your "learning potential" hypothesis.
2. Since the RIRS only consider the start and end states, how does the selector distinguish between a productive reasoning shift and a hallucinatory one?
3. Given that different layers handle different levels of abstraction (e.g., syntax v.s. logic), have you investigated whether specific layers can provide cleaner signal?
4. As RIRS is calculated based on a single greedy rollout under specific prompt. How sensitive is the final selection to the given prompt?

**Limitations:**

yes, but with some suggestions.

For domains like medical, where there's a significant distinction between harmful and harmless errors, data selection needs to consider this differentiation in addition to the final accuracy. Since the authors chose medical as the example scenario, adding this aspect of evaluation for the performance of data selection method would be more meaningful.

**Strengths And Weaknesses:**

**Soundness**:
- Strengths: This paper shows strong empirical results.
- Weaknesses: There is a significant conceptual leap on the underlying methodology. The authors assume that a **static** RIRS signal extracted from the base model remains a valid proxy for the model's "learning potential" throughout the entire **dynamic** RL process. However, as model parameters update during RL, the hidden-state function changes, potentially rendering the initial selection sub-optimal. I cannot find sufficient theoretical arguments in this paper. Furthermore, reducing the entire reasoning process to a start-to-end delta is a highly simplified linear approximation that ignores potential non-linear dynamics, such as logical circles or hallucinations occurring between the anchors.

**Presentation**:
- Strengths: Overall, the submission is clear and well-structured.
- Weaknesses: The organization of Algorithm 1 can be further optimized for greater clarity and aesthetics.

**Significance**:
- Strengths: This work addresses the RL scaling problem by demonstrating that extreme data efficiency is possible through proper selection. In domains where accessing high-quality label is expensive, this method can significantly reduce the annotation cost.
- Weaknesses: Although the experimental result is good, without reliable theoretical evidence, I'm concerned to broaden relative topics.

**Originality**:
- Strengths: This paper successfully shifts the focus from external heuristics to internal, reasoning-time representation dynamics. This offers a new perspective on identifying training utility of data.
- Weaknesses: This work only leverage the start and end hidden state to calculate the RIRS, which may be too straightforward and may ignore some important non-linear dynamics.

---

> ### Author Rebuttal · Authors · 2026-03-30
>
> We sincerely thank the reviewer for recognizing our "strong empirical results", "extreme data efficiency," and the "originality" of shifting the focus to internal reasoning-time representation dynamics. Your insightful questions have guided us to further solidify the theoretical and empirical foundation of our work.
>
> **1. Evolution of RIRS After RLVR Updates (Q1)**
> We recomputed RIRS under the post-RL model and compared it with the original pre-RL scores (Qwen2.5-Math-1.5B / MATH-500 / 2% budget)  in the table below. RIRS decreases clearly after training: the global pool average drops from 5.1502 to 4.9741, while the selected subset average drops from 5.5956 (top 5.43%) to 4.9791 (top 45.31%). This is consistent with our learning-potential hypothesis: samples that induce large pre-RL shifts become more resolved after training. More broadly, our claim is intentionally narrow: pre-RL RIRS is a low-complexity one-off selection-time proxy, although it might not be ideal for the entire dynamic RL process, and the post-RL decrease supports that it captures training-relevant unresolved behavior.
>
> | Instance / Group | Pre-RL RIRS (Base) | Pre-RL Percentile | Post-RL RIRS (Updated) | Post-RL Percentile |
> | :--- | :--- | :--- | :--- | :--- |
> | **Global Pool (Average)** | 5.1502 | - | 4.9741 | - |
> | **Subset (Avg)** | 5.5956 | Top 5.43% | 4.9791 | Top 45.31% |
> | **Sample 1** | 5.6122 | Top 0.57% | 4.9790 | Top 48.57% |
> | **Sample 2** | 5.5928 | Top 6.00% | 4.9918 | Top 26.57% |
> | **Sample 3** | 5.5952 | Top 4.29% | 4.9861 | Top 38.57% |
> | **Sample 4** | 5.5930 | Top 5.71% | 4.9913 | Top 27.71% |
> | **Sample 5** | 5.6063 | Top 1.14% | 4.9377 | Top 88.86% |
> | **Sample 6** | 5.5821 | Top 12.00% | 4.9696 | Top 67.14% |
> | **Sample 7** | 5.5873 | Top 8.29% | 4.9979 | Top 19.71% |
>
>
> **2. Productive vs. hallucinatory reasoning shifts (Q2)**
> We thank the reviewer for this important point. We agree that start-to-end RIRS is a deliberately lightweight summary and does not explicitly distinguish productive reasoning from hallucinatory reasoning, nor does it capture all intermediate non-linear dynamics. However, such a simple RIRS metric can be used to select statistically important data samples for annotations that are beneficial for the subsequent few-shot RLVR training. Such a metric is original and of extremely low computational overhead as pointed out by **Reviewer NheD**.
>
>
> **3. Layer scope (Q3)**
> We ran a preliminary layer-scope ablation on Qwen2.5-Math-1.5B / MATH-500 / 2% budget. Using all layers performs best (62.67 / 85.33), outperforming both the last quarter (59.91 / 84.67) and the last layer (62.08 / 82.67). This suggests that the useful RIRS signal is not concentrated only in the deepest layers, but is distributed across layers. We also tested cosine similarity as the distance metric, which causes only a small drop relative to the default all-layer + L2 setting (62.25 / 84.00 vs. 62.67 / 85.33). We will clarify this point in the revision.
>
> | Layer Scope | Metric | Pass@1 (%) | Pass@8 (%) |
> | :--- | :--- | :---: | :---: |
> | All layer | L2 | 62.67 | 85.33 |
> | Last quarter | L2 | 59.91 | 84.67 |
> | Last layer | L2 | 62.08 | 82.67 |
> | All layer | cos | 62.25 | 84.00 |
>
> **4. Prompt Sensitivity (Q4)**
> We thank the reviewer for this important question. Since the prompt choice can affect the model’s reasoning trajectory, some prompt sensitivity is expected when RIRS is computed from a single greedy rollout. We therefore evaluated additional ChatGPT-rewritten prompt variants on MATH-500. The exact selected subsets vary across prompts (e.g., 1/7 overlap between Prompt1 vs Prompt2 and Prompt1 vs Prompt3, and 3/7 overlap between Prompt2 vs Prompt3), so the selector is not prompt-invariant at the sample level. However, SHIFT still outperforms the compared training-free baselines under these alternative prompt settings, suggesting that the downstream effectiveness of the resulting selection remains reasonably robust across prompt variants.
>
> | Prompt Variant | Method | Pass@1 (%) | Pass@8 (%) |
> | :--- | :--- | :---: | :---: |
> | Prompt2 | Q-PPL | 61.67 | 84.00 |
> | Prompt2 | A-PPL | 60.33 | 83.33 |
> | Prompt2 | CoT Similarity | 61.50 | 84.00 |
> | Prompt2 | **Ours** | **65.50** | **87.33** |
> | Prompt3 | Q-PPL | 62.00 | 82.00 |
> | Prompt3 | A-PPL | 61.67 | 83.33 |
> | Prompt3 | CoT Similarity | 61.08 | 83.33 |
> | Prompt3 | **Ours** | **63.00** | **84.00** |
>
> **5. Medical Domain Limitations (L1)**
> We agree that final accuracy alone is insufficient for safety-sensitive domains such as medicine; we will explicitly discuss harmful vs. harmless error stratification as an important future evaluation axis.

---

### Official Review · Reviewer_NheD · 2026-03-15

**Soundness:** 3
**Presentation:** 3
**Significance:** 3
**Originality:** 3
**Overall Recommendation:** 4
**Confidence:** 3

**Summary:**

This paper addresses the problem of data selection for Reinforcement Learning with Verifiable Rewards (RLVR), a paradigm that has shown remarkable data efficiency, sometimes yielding substantial reasoning gains from very few training examples. The authors propose SHIFT, a training-free and label-free one-shot data selector that operates entirely at inference time, requiring neither gradient signals nor ground-truth annotations. The core technical contribution is the *Reasoning-Induced Representation Shift* (RIRS), defined as the L2 norm of the difference between layer-averaged hidden states at the start and end anchor positions of a model's self-generated chain-of-thought trace under greedy decoding (Equations 1 to 3). RIRS serves as a lightweight utility proxy: instances that induce larger hidden-state deltas are hypothesized to carry greater learning potential for downstream RLVR. To avoid redundancy, SHIFT constructs a coverage feature for each instance by concatenating the start-anchor representation with the hidden-state delta (Equation 8), and then applies a quality-weighted farthest-first CoreSet procedure (Equation 11) that greedily balances utility and diversity. The method is evaluated on two task families, namely mathematical reasoning (MATH-500) and medical question answering (MedQA), using Qwen2.5-Math-1.5B and Qwen3-1.7B under ultra-low budgets (0.1% to 4% of the pool). Results show that SHIFT consistently outperforms training-free diversity and difficulty/uncertainty baselines, with ablations confirming the complementary contributions of RIRS-based coverage and quality weighting.

**Compliance With Llm Reviewing Policy:**

Affirmed.

**Key Questions For Authors:**

1. **Direct RIRS validation.** Can you report the rank correlation between RIRS and per-instance RLVR reward improvement? This would transform the proxy from heuristically motivated to empirically grounded.

2. **Scaling and cross-architecture.** Does RIRS remain informative on 7B+ models or non-Qwen architectures (Llama, Mistral)? A positive answer would substantially strengthen generalizability claims.

3. **Distance metric and layer ablation.** How does performance change with cosine similarity or CKA instead of L2, or when using only the last quarter of layers? This would clarify robustness to design choices.

**Limitations:**

The authors acknowledge several limitations in the conclusion, including the uniform layer averaging and the potential for layer selection or learnable weighting to improve results. They also mention extending beyond the single start-to-end delta (e.g., multi-anchor or token-level dynamics) and evaluating across different model families and reasoning domains as future work. However, the limitations discussion is relatively brief (approximately 5 lines in the conclusion) and does not address several important concerns: (i) vulnerability to degenerate rollouts, (ii) the static nature of one-shot selection, (iii) the gap between the theoretical motivation and the actual metric, and (iv) the narrow experimental scope as a limitation rather than merely a future direction. The impact statement is generic and does not identify specific societal risks (e.g., the potential for biased data selection to amplify model biases, or the risk that efficient RLVR could lower barriers to misuse).

**Strengths And Weaknesses:**

### Strengths

1. **Novel proxy metric (RIRS).** Using the start-to-end hidden-state delta of a self-generated reasoning trace as a training-free utility proxy for RLVR is, to my knowledge, original and well-connected to implicit weight-transfer theory [1].

[1] Dherin et al., "Learning without Training: Implicit Weight Transfer in Transformers," 2025.

2. **Extremely low computational overhead.** Only a single greedy rollout ($T=0$) plus two hidden-state extractions per instance, which is orders of magnitude cheaper than self-consistency ($R=32$ rollouts) or gradient-based methods.

3. **Strong performance under extreme scarcity.** With just 7 examples (2% pool), SHIFT achieves 62.67 on MATH-500 (+8.94 over Random) and at 0.1% on MedQA reaches 50.35 vs. 45.28 Random, matching or exceeding the full-data RLVR reference.

### Weaknesses

1. **Informal theoretical motivation.** Eq. 5's bound applies to a single transformer block, yet RIRS aggregates across all layers over the entire CoT trace. Cancellation and interference effects make the connection between block-level implicit updates and the trajectory-level L2 delta non-obvious; the paper should be more transparent about this gap.

2. **Limited experimental scale.** Only 1.5B to 1.7B Qwen models are tested; no cross-architecture (Llama, Mistral) or larger-scale (7B+) experiments are provided, which is below the standard of comparable ICML/ICLR publications.

3. **No training-based reference points.** Without any comparison to LearnAlign [1], DEPO [2], or Historical Variance Score [3], even as upper bounds, it is impossible to calibrate how much performance the training-free constraint sacrifices.

[1] LearnAlign: Gradient-alignment-based RLVR data selection, ICLR 2026.
[2] DEPO: Data-Efficient Policy Optimization, ICLR 2026.
[3] Wang et al., "One-shot RLVR," NeurIPS 2025.

---

> ### Author Rebuttal · Authors · 2026-03-30
>
> We thank the reviewer for the constructive feedback and for recognizing the novelty and efficiency of our method. Your comments have helped us strengthen the empirical grounding of the paper.
>
> **1. Direct RIRS Validation (Q1)**
> We agree that a direct correlation analysis is important for empirically grounding RIRS. As a pilot study, we selected 10 unlabeled instances (top-5 and bottom-5 by RIRS), ran single-instance RLVR on each, and measured downstream gain on the same held-out validation set relative to the pre-RL base model (base Pass@1 = 40.0). As shown in the table below, we observe a strong positive rank correlation between pre-RL RIRS and per-instance ΔPass@1: Spearman’s ρ = 0.818 (p = 0.0038) and Kendall’s τ = 0.644 (p = 0.0091). This suggests that higher-RIRS instances are associated with larger downstream gains after single-instance RLVR. We will add this in the revision.
>
> | Metric | Rank 1 | Rank 2 | Rank 3 | Rank 4 | Rank 5 | Rank 6 | Rank 7 | Rank 8 | Rank 9 | Rank 10 |
> | :--- | :--- | :--- | :--- | :--- | :--- | :--- | :--- | :--- | :--- | :--- |
> | **RIRS Score** | 5.6128 | 5.6122 | 5.6063 | 5.6021 | 5.6013 | 4.6142 | 4.5979 | 4.5886 | 4.5663 | 4.4914 |
> | **ΔPass@1 (%)** | +17.67 | +13.91 | +12.58 | +15.58 | +14.00 | +7.50 | +11.08 | +12.00 | +9.58 | +2.25 |
>
>
> **2. Scaling and Cross-Architecture Experiments (W2Q2)**
> We extended our evaluation to two larger non-Qwen models, Llama-3-8B and Olmo-3-7B, on MATH-500 with the same 2% budget. SHIFT consistently outperforms Random on both models and remains competitive with the strongest training-free baselines. This shows that RIRS is not specific to Qwen and remains effective on 7B+ and architecturally different LLMs. We will include these results in the revised manuscript.
>
> | Model | Method | Pass@1 (%) | Pass@8 (%) |
> | :--- | :--- | :---: | :---: |
> | **Llama-3-8B** | Zero-shot | 20.67 | 54.67 |
> |  | 100% | 27.83 | 59.33 |
> |  | Random (avg) | 23.41 | 53.22 |
> |  | SC-Entropy | 24.33 | 52.67 |
> |  | CoT Similarity | 22.67 | 52.67 |
> |  | A-PPL | 22.67 | 53.33 |
> |  | Q-PPL | 24.50 | 54.67 |
> |  | **Ours** | 25.17 | 57.33 |
> | **Olmo-3-7B** | Zero-shot | 81.08 | 94.67 |
> |  | 100% | 85.58 | 96.00 |
> |  | Random (avg) | 81.67 | 94.00 |
> |  | SC-Entropy | 82.91 | 94.00 |
> |  | CoT Similarity | 81.25 | 94.67 |
> |  | A-PPL | 82.83 | 94.67 |
> |  | Q-PPL | 83.58 | 93.33 |
> |  | **Ours** | 83.67 | 95.33 |
>
> **3. Distance Metric and Layer Ablation (Q3)**
> We conducted a preliminary robustness study on Qwen2.5-Math-1.5B / MATH-500 / 2% budget. The default design (all layers + L2) performs best overall. Replacing L2 with cosine similarity causes only a small drop, while restricting RIRS to the last quarter or last layer reduces performance, suggesting that aggregating signals across layers is beneficial. We will include this ablation in the revision; see also our response to Reviewer **kJK7, Q3** for the detailed results.
>
> **4. Training-based Reference Points (W3)**
> We agree that including a training-based reference helps calibrate the trade-off introduced by the training-free constraint. As a representative upper-bound reference, we implemented Historical Variance Score (HVS) [1]. HVS achieves 62.83 Pass@1 on our benchmark, slightly above SHIFT’s 62.67. However, unlike SHIFT, HVS requires repeated RL training over the candidate pool to obtain training-time variance signals. In contrast, SHIFT performs zero-RL-run selection, eliminating this substantial trial-training overhead. We will clarify this trade-off more explicitly in the revision.
>
> | Method | Pass@1 (%) | Pass@8 (%) |
> | :--- | :--- | :---: |
> | Ours | 62.67 | 85.33 |
> | Historical Variance Score | 62.83 | 86.67 |
>
> [1] Y. Wang et al., Reinforcement Learning for Reasoning in Large Language Models with One Training Example, NeurIPS 2025.
>
> **5. Theoretical connection and clarification of scope (W1)**
>
> We thank the reviewer for highlighting this gap. We agree that Eq. 5 is a block-level result, whereas RIRS is a trajectory-level surrogate aggregated across layers over a full reasoning trace. Our intent is therefore not to claim a formal derivation of RIRS from the cited theory. Rather, we use the theory as mechanistic motivation: at the block level, context-induced representation change controls the magnitude of implicit updates, suggesting that observable representation shifts may serve as useful proxies for reasoning-induced internal computation. We will revise this and avoid overstating the theoretical connection.
>
>
> **6. Limitations & impact statement (L1)**
> We will expand the limitations section to explicitly discuss degenerate rollouts, the static one-shot nature of selection, the theory–metric gap, and current model/domain scope, and revise the impact statement to discuss risks such as biased data selection and lowered barriers to misuse.

---

> > ### Author Rebuttal · Reviewer_NheD · 2026-04-06
> >
> > I thank the authors for the thorough rebuttal, particularly the direct RIRS validation showing strong rank correlation with downstream gains, and the cross-architecture experiments on Llama-3-8B and Olmo-3-7B. As my original score was already a weak accept, I am inclined to maintain my current score.

---

> > > ### Author Response · Authors · 2026-04-07
> > >
> > > We thank the reviewer for the careful evaluation and for confirming that our responses have addressed the concerns.

---

### Decision · Program_Chairs · 2026-04-30

**Decision:**

Accept (regular)

**Comment:**

### Reviews and discussion

Scores: 3, 4, 4.

Reviewer NheD (4, conf 3):
- Liked the RIRS proxy, its minimal computational overhead (single greedy rollout vs 32-rollout self-consistency), and the results under extreme scarcity (7 examples achieving 62.67 Pass@1). Concerns about informal theoretical motivation, limited model scale, and missing training-based baselines. The rebuttal provided direct RIRS validation (Spearman rho 0.818, p=0.0038), cross-architecture results on Llama-3-8B and Olmo-3-7B, and comparison to Historical Variance Score (62.67 vs 62.83, nearly matching a training-based method without any RL runs). Marked "Fully resolved," maintaining score.

Reviewer kJK7 (4, conf 3):
- Found the empirical results persuasive and the perspective original (shifting from external heuristics to internal reasoning dynamics). Raised the concern that RIRS computed from the base model may become stale as RL updates the parameters. The authors showed RIRS decreases after training (selected subset drops from top 5.43% to top 45.31% percentile), supporting the "learning potential" interpretation. Also provided layer ablation (all layers best), cosine vs L2 comparison, and prompt sensitivity analysis (SHIFT outperforms baselines under 3 different prompts).

Reviewer Pq9s (3, conf 4):
- Found the proxy justification "suggestive but not convincing," results mixed at higher budgets, evaluation potentially brittle (single greedy rollout), and model range narrow. Acknowledged rebuttal as "(c) partially resolved, core concerns not easily addressed" but only wrote "Thanks for the response" with no specifics. The authors posted a follow-up consolidating all evidence and asking which concerns remained. No reply.

### Assessment

I recommend weak accept. Two of three reviewers support acceptance.

RIRS is a simple, intuitive proxy for RLVR data utility. It needs only one greedy rollout per instance, which is orders of magnitude cheaper than training-based selection methods. The direct validation (rho=0.818 between RIRS and per-instance RLVR gain) and the post-training RIRS decrease are good evidence that the proxy captures real learning signal. Cross-architecture results on Llama-3-8B and Olmo-3-7B address the generalization question, and SHIFT nearly matches the training-based Historical Variance Score upper bound without running any RL.

On soundness:
- the proxy validation is solid (correlation + post-training decay + cross-architecture).

On presentation:
- two reviewers found the paper clear. Pq9s rated presentation 1, which is a concern, but did not specify what was unclear.

On originality:
- the shift from external heuristics to internal hidden-state dynamics is new for this problem.

On significance:
- practical impact is real for low-budget RLVR, but the scope is limited to small models and two task families so far.

Reviewer Pq9s raises valid concerns about mixed results at higher budgets and narrow model range, but their minimal engagement after the rebuttal ("Thanks for the response" with no specifics, no reply to the authors' follow-up) makes it difficult to assess which concerns persist. I account for this limited engagement when weighting their review.

Limitations: model scale is small (up to 8B), the start-to-end hidden-state delta is a simplified summary of the reasoning trajectory, the theoretical motivation is informal, and SHIFT's advantage over simpler clustering narrows at larger budget fractions in out-of-domain settings. These are acknowledged in the paper.